# Data-driven method to infer the seizure propagation patterns in an epileptic brain from intracranial electroencephalography

Viktor Sip[1], Meysam Hashemi[1], Anirudh N. Vattikonda[1], Marmaduke M. Woodman[1], Huifang Wang[1], Julia Scholly[2,3], Samuel Medina Villalon[1,3], Maxime Guye[2,4], Fabrice Bartolomei[1,3], Viktor K. Jirsa[1]*

**1** Aix Marseille Univ, INSERM, INS, Inst Neurosci Syst, Marseille, France, **2** Assistance Publique - Hôpitaux de Marseille, Hôpital de la Timone, CEMEREM, Pôle d'Imagerie Médicale, CHU, Marseille, France, **3** Assistance Publique - Hôpitaux de Marseille, Hôpital de la Timone, Service de Neurophysiologie Clinique, CHU, Marseille, France, **4** Aix Marseille Univ, CNRS, CRMBM, Marseille, France

* viktor.jirsa@univ-amu.fr

**Data Availability Statement:** The patient data sets cannot be made publicly available due to the data protection concerns regarding potentially

## Abstract

Surgical interventions in epileptic patients aimed at the removal of the epileptogenic zone have success rates at only 60-70%. This failure can be partly attributed to the insufficient spatial sampling by the implanted intracranial electrodes during the clinical evaluation, leading to an incomplete picture of spatio-temporal seizure organization in the regions that are not directly observed. Utilizing the partial observations of the seizure spreading through the brain network, complemented by the assumption that the epileptic seizures spread along the structural connections, we infer if and when are the unobserved regions recruited in the seizure. To this end we introduce a data-driven model of seizure recruitment and propagation across a weighted network, which we invert using the Bayesian inference framework. Using a leave-one-out cross-validation scheme on a cohort of 45 patients we demonstrate that the method can improve the predictions of the states of the unobserved regions compared to an empirical estimate that does not use the structural information, yet it is on the same level as the estimate that takes the structure into account. Furthermore, a comparison with the performed surgical resection and the surgery outcome indicates a link between the inferred excitable regions and the actual epileptogenic zone. The results emphasize the importance of the structural connectome in the large-scale spatio-temporal organization of epileptic seizures and introduce a novel way to integrate the patient-specific connectome and intracranial seizure recordings in a whole-brain computational model of seizure spread.

## Author summary

The electrical activity of the brain during an epileptic seizure can be observed with intracranial EEG, that is electrodes implanted in the patient's brain. However, due to the practical constraints only selected brain regions can be implanted, which brings a risk that the abnormal electrical activity in some non-implanted regions is hidden from the observers.

identifying and sensitive patient information. Interested researchers may access the data sets by contacting Clinical Data Manager Aurélie Ponz (aurelie.ponz@univ-amu.fr) at the Institut de Neurosciences des Systèmes, Aix-Marseille Université. The code is available at https://github.com/ins-amu/ddmisp.

**Funding:** This work was funded by the French National Research Agency (ANR) as part of the second "Investissements d'Avenir" program, ANR-17-RHUS-0004, EPINOV (https://anr.fr) to VJ, FB, and MG, by European Union's Horizon 2020 Framework Programme for Research and Innovation under the Specific Grant Agreement No. 785907 and 945539, Human Brain Project SGA2 and SGA3 (https://ec.europa.eu/programmes/horizon2020) to VJ, by European Union's Horizon 2020 Framework Programme for Research and Innovation under the Specific Grant Agreement and No. 826421, VirtualBrainCloud (https://ec.europa.eu/programmes/horizon2020) to VJ, and by SATT Sud-Est, 827-SA-16-UAM (https://www.sattse.com) to VJ, FB, and MG. The funders had no role in study design, data collection and analysis, decision to publish, or preparation of the manuscript.

**Competing interests:** The authors have declared that no competing interests exist.

In this work we introduce a method to infer what is happening in the unobserved parts based on the incomplete observations of the epileptic seizure. The method relies on the assumption that the seizure spreads along the white-matter structural connections, and finds the explanation of the whole-brain seizure spread consistent with the data. The structural connectome can be estimated from diffusion-weighted imaging for an individual patient, therefore this way the patient-specific structural connectome is utilized to better analyze the patients' seizure recordings.

# 1 Introduction

A possible treatment for patients with drug-resistant epilepsy is a surgical intervention aimed at the removal of the suspected epileptogenic zone (EZ), i.e. the brain region responsible for the initiation of the seizures whose removal would result in seizure freedom. However, these surgical interventions have success rates in rendering the patients seizure-free at only 60-70% [1, 2]. Why that can be? The current standard in pre-surgical evaluation is the use of either implanted depth electrodes (stereo-electroencephalography, SEEG) or subdural electrode grids [3]. Interpreting the electrographic signals is however not straightforward due to the complex local dynamics and interactions between brain regions [4, 5], and the degree of epileptogenicity of brain structures and the extent of highly epileptogenic tissue might be misestimated. Furthermore, intracranial EEG does not allow for the exploration of the whole brain, and it is biased to the regions suspected to be part of the epileptogenic network based on the non-invasive evaluation. This introduces a risk that the highly epileptogenic tissue is not fully explored by the implantation, leading again to an incomplete resection.

In this work we explore if these issues can be solved by exploiting the role of structural connections in spatio-temporal seizure organization. In healthy brains there exists a link between the structural and functional connectivity obtained from resting state fMRI [6, 7]. Studies indicate that in epileptic brains the structural connectivity is altered [8] as is the structure-function relationship [9], and, importantly for this study, that the long-range white matter connections shape the spread of epileptic seizures [10–12].

The increasing availability of diffusion-weighted magnetic resonance imaging in clinical practice allowed for building patient-specific structural brain networks, which opened the way for network-based computational models of epileptic activity [13]. Several such models appeared in past years with the aim to explore the possibilities of surgical interventions and to predict their outcome [11, 14–17]. Importantly, unlike the models based on the functional connectivity networks derived from interictal and/or ictal intracranial EEG recordings [18–22], the models based on the structural connectivity are not spatially restricted to the implanted brain regions and can simulate the whole brain dynamics. In some of the models, the heterogenity of the node behavior is caused only by the underlying connectivity [18, 19]. In others a spatially heterogeneous parameter representing node excitability is introduced; these models explore the effects of the heterogeneity of local parameters in small synthetic networks [23–25], or model the whole-brain dynamics with the local excitability informed by the patient-specific anatomy [14] or by the clinical hypothesis of the node excitability [11, 15, 17].

While these network models of epilepsy can provide valuable insight into the role of the network in seizure organization, they were designed with forward simulations in mind and might not pose an easy target for model inversion. Here by model inversion we mean the process of finding the model parameters (such as the parameters of the neural masses in network nodes or connection strengths) that can best explain given observations of the network

dynamics. Indeed, most of the model inversion studies related to epileptic seizures so far dealt with a single neural mass, small networks of neural masses, or uncoupled neural masses [26–32]. Only recently have studies on large-scale network model inversion appeared [33], however, inversion of large systems of coupled differential equations can still be prohibitively computationally expensive and requires a careful balance of the model parameterization, choice of the prior distributions of the parameters, and settings of the inversion method.

In this work we approach the problem of seizure propagation differently: instead of attempting to invert pre-existing model of seizure propagation, we introduce a novel model designed with the inversion in mind. As such, the model drops some complexity of the existing models while keeping the core elements of network models of seizure propagation: two possible states of a network node (either a healthy or a seizure state), the role of the network connections in seizure spread, and the role of heterogeneous node excitability. This state-based approach can be reminiscent of the cellular automatons, used also for epilepsy modeling [34]. In exchange for this simplification we obtain a model which can be reliably inverted both in a single-seizure regime to obtain seizure-specific regional excitabilities, or in multi-seizure regime to obtain the optimal hyperparameters of the model shared between seizures. The model is thus data-driven: we provide only its generic form and infer the parameters from the data. Using the Bayesian inference framework, we invert the model using the Markov Chain Monte Carlo (MCMC) method in order to quantify the uncertainty of the estimations.

In the conceptual view adopted here the brain network during a seizure is only incompletely observed, with some regions observed as non-seizing and some as seizing with a specific onset time of seizure activity (Fig 1A). Using a collection of seizures recorded from several patients, patient-specific brain networks, and the introduced model, we perform the model inversion to obtain the hyperparameters of the model and the seizure-specific excitabilities, and to fill in the unknown onset times of hidden nodes (Fig 1B). We validate the method on synthetic, model-generated data, as well as on real data recorded from patients with drug-resistant epilepsies. To do the latter we use two complementary approaches (Fig 1C): The leave-one-out validation tests how well can the method predict the state and onset time of the hidden regions by excluding one observed region from the data, fitting the model without it, and comparing the prediction with the left-out information. The resection validation tests how well can the method predict the surgery outcome using the patient-specific fitted models and the actual resection performed.

## 2 Results

### 2.1 Overview of the method

While the detailed description of the method is given in 4, here we provide a general overview of the method and of the assumptions behind it. At the core of the method is the dynamical model of seizure propagation in the brain network. The model is constructed so it, in the simplest fashion possible, expresses the following assumptions about the seizure propagation:

- The seizure propagates through the brain network, represented as a weighted network $W = \{w_{ij}\} \in \mathbb{R}^{n \times n}$, where $w_{ij}$ stands for the connection strength from region $j$ to region $i$, and $n$ stands for the number of brain regions. We estimate this patient-specific structural connectivity from diffusion-weighted MRI images.

- Each node of the network is at any given point in time either in a healthy state or in a seizure state.

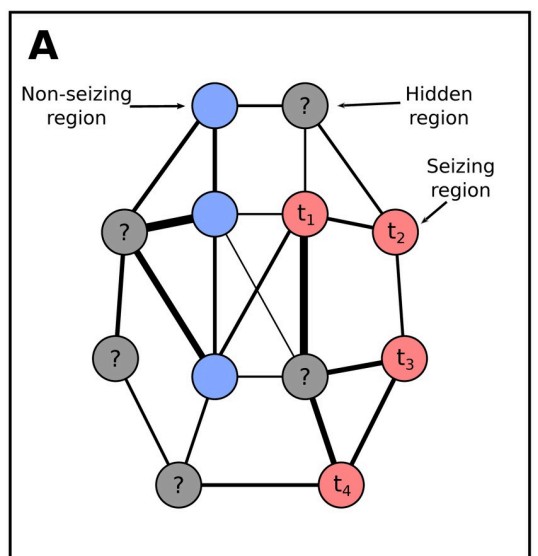

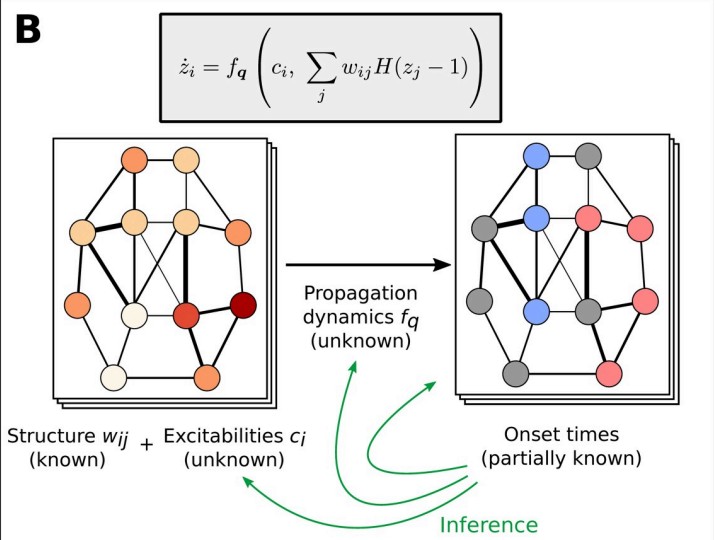

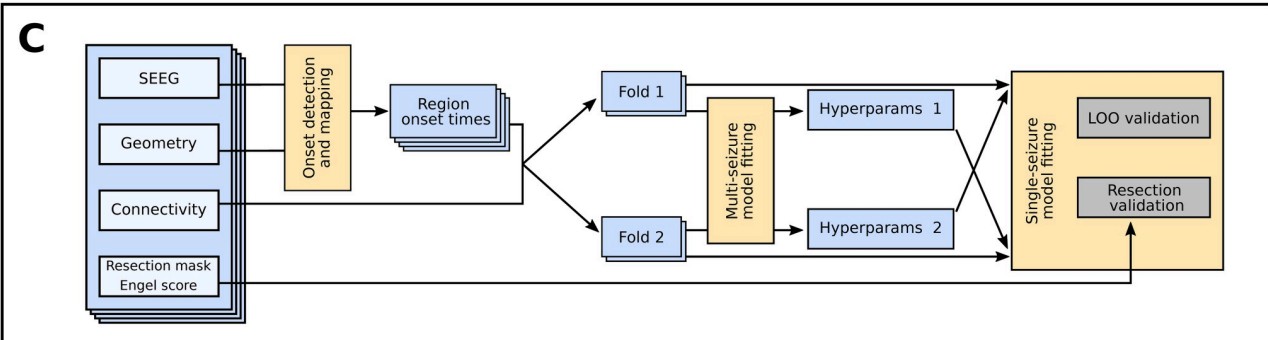

**Fig 1. Main concepts and organization of the work.** (A) Observation of a spreading seizure in the brain network. Due to the sparseness of the implanted electrodes, only some regions are observed; for those we know their state, non-seizing or seizing, and the onset time in the latter case. (B) Generative model of the seizure and the problem of the inference. We assume that the region onset times result from initially unknown propagation dynamics, shared among all seizures, and depend on the known patient-specific network structure and unknown seizure-specific region excitabilities. The goal of the inference is to infer the form of the propagation dynamics and the seizure-specific region excitabilites, and thus also the missing onset times. (C) The workflow used in this study for the validation on real data of 45 subjects. From the SEEG recordings the channel onset times were extracted and mapped onto brain regions. The data set was then divided into two folds, and each was fitted separately with the multi-seizure model to infer the model hyperparameters. The leave-one-out validation and the resection validation were then performed using the single-seizure model with hyperparameters obtained from the other fold.

- The sudden transition of a brain region $i$ from the healthy to the seizure state is driven by continuous changes in the slow variable $z_i$, loosely corresponding to the slow permittivity variable in the Epileptor model [35] or to the usage-dependent exhaustion of inhibition in the model of Liou and colleagues [36]. All regions are initially in a healthy state, and any region starts to seize when its slow variable crosses a given threshold.

- The rate of change of the slow variable of a region $i$ depends only on its inner excitability $c_i$ and the input it receives from the connected seizing regions.

Formally, the dynamical model of the seizure propagation is written as follows:

$$\dot{z}_i \;=\; f_q\left(c_i, \sum_j w_{ij} H(z_j - 1)\right), \quad z_i(0) = 0, \tag{1}$$

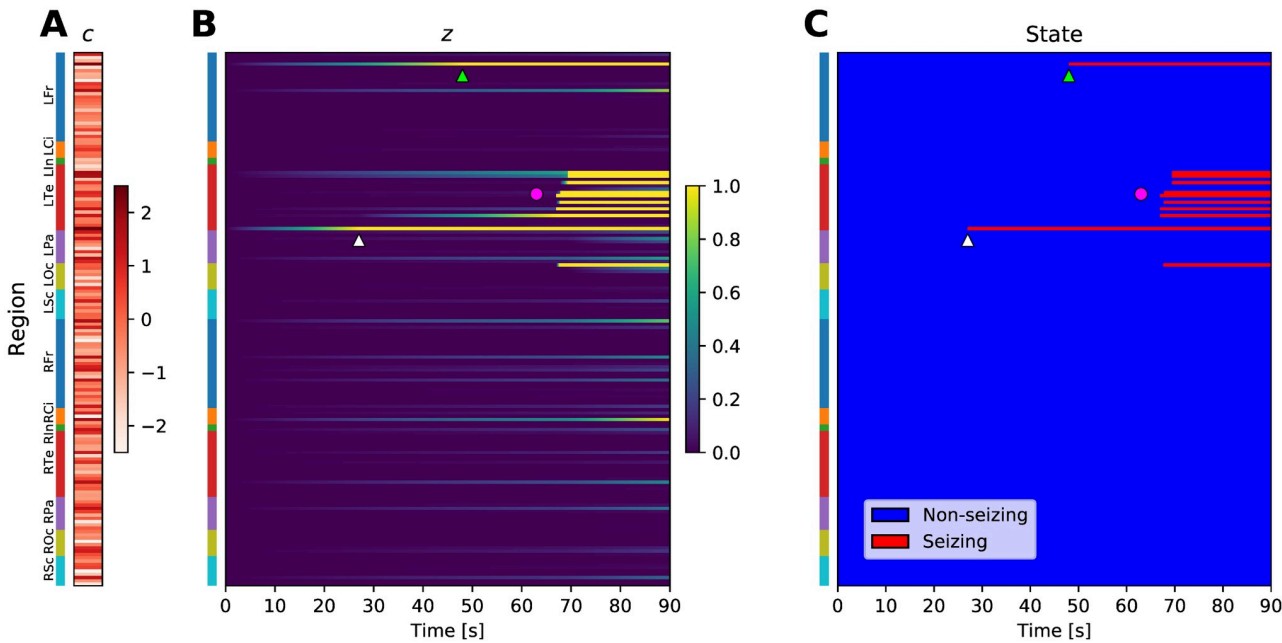

**Fig 2. Example of a simulated seizure in real brain network.** (A) The region excitabilities $c$ are randomly sampled from a standard normal distribution. Brain regions are located on the vertical axis, colored bars indicate the anatomical grouping of the regions. (B) Slow variable $z$, simulated using the sampled excitabilities and *strong coupling* excitation function (see Fig 2 in S1 Text) on a brain network of subject 1. (C) Seizure state of the regions, obtained by thresholding the slow variable at $z = 1$. Due to the elevated excitability of the regions, the seizure starts in the Rhinal cortex in left temporal lobe (white triangle), and with some delay in Pars orbitalis in left frontal lobe (light green triangle). Eventually, large portion of the left temporal lobe is recruited (magenta circle). Abbreviations: Lxx/Rxx—left/right hemisphere, Fr—frontal lobe, Ci—cingulate cortex, In—Insula, Te—temporal lobe, Pa—parietal lobe, Oc—occipital lobe, Sc—subcortical structures. See Table 2 in S1 Text for a full list of regions.

where $z_i$ is the slow variable, $c_i$ is the region excitability, $w_{ij}$ are the connection strengths, and $H(z_j - 1)$ is the Heaviside function, representing the seizure threshold at $z_j = 1$. We call the time point $t_i$ when region $i$ crosses the threshold and starts to seize at *onset time* of region $i$. Finally, the function $f_q$ is the *excitation* function parameterized by the parameter vector $\boldsymbol{q}$.

We require the function $f_q$ to be positive; consequently, all regions will switch to the seizure state in finite time. That is clearly not the case in reality, and we thus associate all regions that start to seize after limit time $t_{\text{lim}}$ with non-seizing regions. This constant expresses the time scale we consider relevant for seizure spread; in this work we set $t_{\text{lim}} = 90$. Associating late-seizing with non-seizing regions might at first seem like a poor approximation of reality, it however serves a crucial purpose for the model inversion: it allows us to avoid a problem of a discrete nature (inferring *if* a region seizes) and replace it with a continuous problem (inferring *when* a region starts to seize), which is better suited for inference using the MCMC methods. This approach can be justified also from the clinical perspective: the primary objective of the method is to understand the relations between the regions that seize early, and the interactions of the late-seizing or non-seizing regions are only of secondary importance.

Given the parameters $\boldsymbol{q}$, the connectome matrix $W$, and the excitabilities $\boldsymbol{c}$, the dynamical model determines the onset times $\boldsymbol{t}$ (Fig 2). The problem we are facing is however the opposite: given the partially observed onset times and the connectome matrices (for multiple seizures and multiple subjects) we want to infer the parameters $\boldsymbol{q}$ of the excitation function, which we assume are shared among all seizures and for all patients, and the seizure-specific excitabilities. This assumption is made for the sake of simplicity; we discuss the limitations of it in Discussion. Once the parameters $\boldsymbol{q}$ and $\boldsymbol{c}$ are inferred, the unknown onset times can be easily

calculated. We perform this inversion in Bayesian inference framework using the statistical model built upon the dynamical model with additional assumptions:

- The propagation dynamics (or, in the model terms, the parameters $q$ of the excitation function) is shared among all seizures. The individual variability in the observed onset times is therefore assumed to be caused only by the seizure-specific region excitabilities $c$ and the patient-specific structural network $W$.

- The excitabilities $c$ are *a priori* assumed to follow the standard normal distribution $N(0, 1)$.

- The onset times are detected inexactly with normally distributed observation error.

These assumptions lead to a hierarchical statistical model: the hyperparameters at the top level are the parameters $q$ shared among all seizures, while at the bottom level are the seizure-specific excitabilities $c$. Intuitively, the goal of the inference is to find such parameterization $q$ of the excitation function so that the observations of all seizures can be explained by excitabilities that are as close to being normally distributed as possible given the constraints posed by the model formulation.

Due to the simplicity of the dynamical model, mathematical statements about the existence and uniqueness of the solution of the inverse problem can be made in case of no observation error (S1 Text/Note on parameter identifiability). In particular, we show that a solution of the inverse problem is guaranteed to exist for any combination of onset times of the seizing nodes. In fact, in the network with $n_{ns}$ non-seizing nodes, $n_{sz}$ seizing nodes, and $n_{hid}$ hidden nodes, there exist infinite amount of solutions, represented by $(n_{ns} - n_{hid})$-dimensional manifold in the parameter space.

Furthermore, the model can be robustly inverted even for large brain networks, as we demonstrate on extensive validation with synthetic data (S1 Text/Validation on synthetic data). We show that the hyperparameters $q$ of the model can be recovered in multiple scenarios, and we illustrate the limits of the method regarding the recovery of the region-specific excitabilities $c$ for both observed and hidden regions. We also evaluate the capacity of the model to discover the epileptogenic zone which is not explored by the implantation, and demonstrate the role of the network connections in this effort (S1 Text/Discovery of epileptogenic zones in synthetic seizures).

## 2.2 Structural connectomes

The imaging protocols differed between the patients, most notably in the number of gradient directions in diffusion-weighted imaging (64 or 200). To assure that this did not affect the study, we compared the two groups in terms of several network properties of the generated structural connectomes. The connectomes did not significantly differ between the two groups ($n_1 = 27$, $n_2 = 23$) in terms of mean node strength (Mann-Whitney $U$ test, $U = 264$, $p = 0.185$), mean clustering coefficient ($U = 245$, $p = 0.103$), or edge density (binarization threshold 0.001, $U = 289$, $p = 0.341$). Difference was observed in terms of network modularity ($U = 175$, $p = 0.004$), but with modest effect size (min/median/max modularity in the two groups 0.394/ 0.451/0.497 and 0.381/0.424/0.498).

## 2.3 Onset time detection and mapping

First step in applying the model to real patient data is the detection of the onset times of seizure activity in the recorded SEEG signals (Fig 3A) and the subsequent mapping of the channel onset times to the brain regions defined by the brain parcellation (Fig 3B). In our cohort of 50

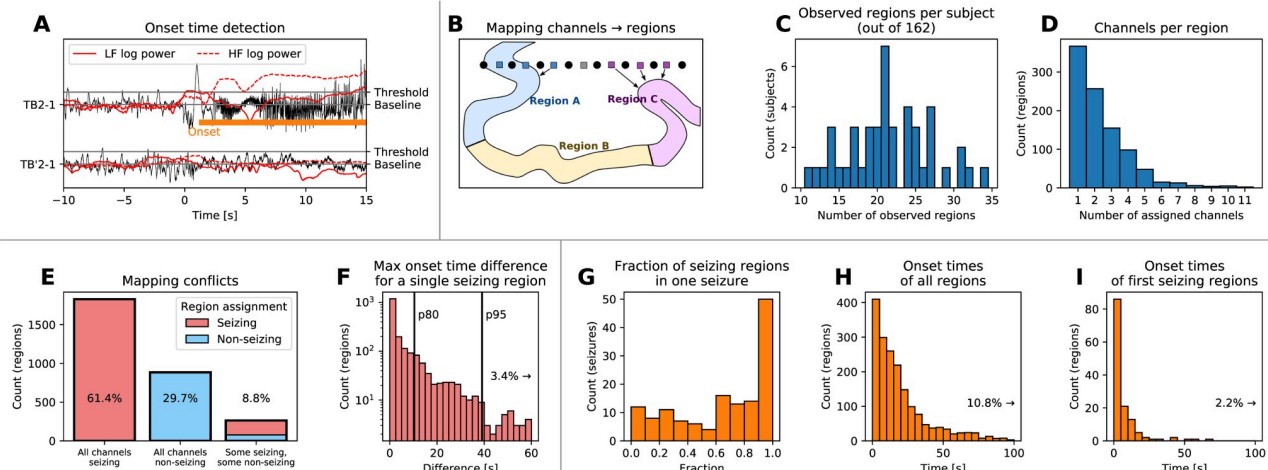

**Fig 3. Onset time detection and mapping to brain regions.** (A) Example of the onset time detection. The onset time on a bipolar SEEG channel is detected by computing the power in lower and high frequencies, normalizing it to preictal levels, and detecting when the power increases above a given threshold. This results in marking a channel as seizing (with detected onset time, upper trace) or non-seizing (lower trace). (B) Channels are assigned to brain regions based on their physical distance. If multiple channels are assigned to the same region, the seizing or non-seizing status is determined based on the majority of channels. If the region is seizing, the region onset time is defined as the median of all assigned onset times, taking the non-seizing regions into account as well with onset time equal to infinity. (C) Histogram of observed regions among all patients in the study. (D) Histogram of assigned SEEG channels per observed region. (E) Histogram of regions based on the seizing state of the assigned channels, indicating where the channel to region mapping leads to a conflict in the region seizing state. Ideally, there would be no regions with some seizing and some non-seizing assigned channels. (F) Histogram of differences between the earliest and latest onset time of assigned channels. Vertical lines indicate the 80th and 95th percentile. (G-H) Results of the detection and mapping. (G) Histogram of the fractions of the seizing regions for all seizures. (H) Difference of the detected onset times of all seizing regions from the clinically marked seizure onset. (I) Difference of the detected onset times of the first seizing region of every seizure from the clinically marked seizure onset.

patients, 45 of them had at least one seizure satisfying the inclusion criteria (total 141 seizures) to which the onset detection was applied. Among them, there was between ten to thirty-five regions observed (i.e. with at least one assigned channel) out of 162 regions in the brain parcellation (Fig 3C); these observed regions have predominantly less than four channels assigned (Fig 3D). The channel to region mapping may lead to one region having multiple channels assigned with different seizing/non-seizing status or different onset times. Such occurrence indicates that the parcellation is not sufficiently fine to properly capture the spatio-temporal dynamics of the seizure, however, we observe that the state of over 90% of the observed regions is unambiguous (Fig 3E). In the remaining cases, the seizing or non-seizing status is determined based on the majority of channels. Furthermore, the difference on the earliest and latest assigned onset time is below ten seconds for 80% of the seizing regions (Fig 3F).

Fig 3G–3I shows the final results of the detection and mapping procedure on the patient data. Seizures where very few, some, or all of the observed regions are seizing are present (Fig 3G). In four seizures out of 141, no seizing regions were detected; these were excluded from further analysis, leaving total of 137 seizures from 44 subjects. Seizures where close to all observed regions are seizing are more represented, possibly reflecting the bias in the SEEG implantation towards the regions where the seizure activity is expected. The frequency of occurrence of detected onset times decays with increasing delay from the clinically marked seizure onset (Fig 3H). In the majority of the seizures, the earliest seizure onset follows the clinically marked seizure onset with little delay (Fig 3I).

Inevitably, the results of the onset time detection (and of the subsequent analysis) depend on the parameters of the detection method, most notably on the signal power threshold that determines the seizing state. On Fig 5 in S1 Text we analyze the influence of this choice.

Unfortunately, the results are smoothly dependent on the threshold value, and no clear value of the threshold presents itself via a plateau or a sharp drop. We continue with the (to an extent arbitrarily) chosen threshold $\delta = 5$, however, we return to the question of the binary distinction of seizing and non-seizing state in Discussion.

## 2.4 Hyperparameter learning

Next, we used the cohort data to infer the hyperparameters of the model. To avoid a data reuse in the subsequent leave-one-out validation, we divided the cohort into two folds of equal size (22 patients each), and the multi-seizure model was fitted twice, independently for each fold. That led to two sets of estimated parameters. At most two seizures per single patient were used for the fitting to avoid biasing the model towards the patients with more recorded seizures.

Fig 4A shows the inferred posterior distribution for the model hyperparameters, together with two measures of convergence: split-chain scale reduction factor $\hat{R}$ and number of effective samples $N_{eff}$ [37]. The $\hat{R}$ value indicates how well the Markov chains mix, and loosely it is defined as the ratio of within-chain and between-chains parameter variance. The value of 1 indicates perfect mixing, and higher values point towards the chains not mixing well. Number of effective samples $N_{eff}$ is the estimate of the number of independent samples with the same estimation power as the obtained autocorrelated samples. Here in some cases the $\hat{R}$ values lie slightly outside of the generally recommended range $\hat{R} < 1.1$ indicating imperfect mixing of the MCMC chains; we consider it acceptable given the practical limitations of the required time for the computations and considering that the hyperparameter posterior distribution will be reduced to point estimate for the second step of fitting the individual seizures. For the same reason we also do not consider the low number of effective samples problematic.

Even though the inferred parameter posterior distribution for both folds do not overlap perfectly, they lie in the same region of parameter space (Fig 4A in the main text and Fig 6 in

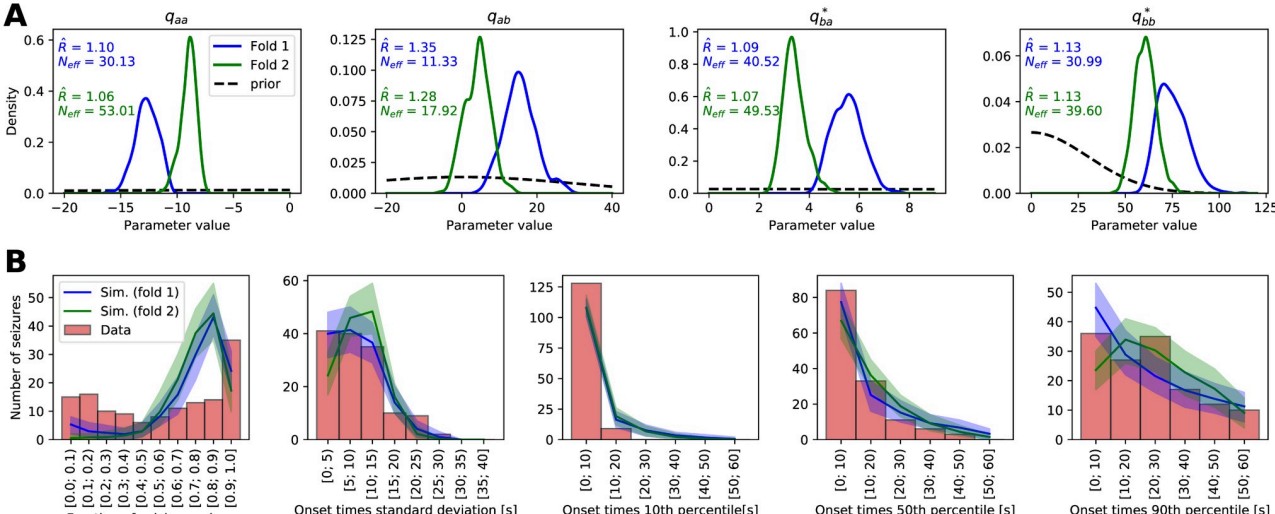

**Fig 4. Results of the hyperparameter learning and posterior predictive checks.** (A) Posterior (full lines) and prior (black dashed lines) distributions of the hyperparameters for the models fitted with two data folds. Hyperparameters $\boldsymbol{q} = (q_{aa}, q_{ab}, q_{ba}^*, q_{bb}^*)$ parameterize the right-hand side function $f_{\boldsymbol{q}}$ of the dynamical model of seizure propagation (1). The text shows the split-chain scale reduction factor $\hat{R}$ and number of effective samples $N_{eff}$. Note the different x and y ranges of the panels. (B) Results of the posterior predictive checks with the fitted models. In all panels, the red histogram shows the properties of the real seizure ensemble, while the solid lines show the mean of the hundred ensembles of simulated seizures and the shaded areas indicate the 5 to 95 percentile range. The panels show, left to right, distribution of the fraction of the seizing regions in one seizure, standard deviation of the onset times of seizing regions, and 10th, 50th, and 90th percentile of the onset times of seizing regions.

S1 Text). The two folds contain data from entirely different patients, these results thus indicate that some common features of the seizures are indeed extracted from the data and that the models are not overfitted to any specific data set.

If the model is sufficiently flexible and fitted well, it should be able to generate synthetic seizures resembling the real ones. Testing if it does is the goal of posterior predictive checks. We took the two hyperparameter sets obtained by fitting the two folds, we randomly drew the excitabilities from the prior standard normal distribution, and we simulated an ensemble of 137 seizures—same number as of the real seizures in the data set. We repeated this process one hundred times to obtain hundred ensembles of simulated seizures for both folds and then we compared the statistical properties of the synthetic seizure ensembles with the ensemble of real seizures (Fig 4B). We chose five statistics to evaluate the similarity between real and simulated seizures: Fraction of the seizing regions in one seizure, and standard deviation and 10th, 50th, and 90th percentile of the onset times of the seizing regions. For a well fitted and sufficiently flexible model, the statistics of the synthetic seizures should overlap with the statistics of the real seizures. Furthermore, the statistics of the simulations from two folds should overlap, the opposite would point to an overfitting to the subsampled data set. The results reveal that the two models from two folds indeed produce statistically similar seizures, indicating again that some common features of seizure dynamics were extracted from the data and the models were not overfitted. However the match with the real seizures is not ideal: the models generate less seizures with only few regions recruited, while overpredicting the number of seizures with majority (but not all) of the regions recruited (Fig 4B, leftmost panel). That might indicate that the model is not sufficiently flexible to reliably reproduce the seizures that do not spread to the whole brain. Other panels in Fig 4B nevertheless show that the statistics of the onset times of seizing regions are well reproduced: the model is sufficient in this aspect.

## 2.5 Single-seizure inference examples

We present four examples of the inference results to illustrate the working of the method. Fig 5 depicts a seizure where the inferred extent of recruited regions is spatially restricted and where the existence of hidden epileptogenic zone is predicted. In this specific case the resective surgery did not result in seizure freedom, it is thus possible that the epileptogenic zone was indeed not explored by the implantation. In contrast, Fig 7 in S1 Text shows a temporal lobe seizure where the inference results mirror the observations, i.e. no early involvement of any non-observed regions is predicted, and the inferred epileptogenic regions coincide with the observed early seizing regions.

An example of a more spatially extended seizure is shown on Fig 8 in S1 Text. The seizure is inferred to start in the right hippocampus before spreading to the majority of the right hemisphere and, eventually, also to the left hemisphere. Despite the inferred involvement of large portion of the brain, the inference points clearly to suspected epileptogenic zone due to the observed early involvement of the right hippocampus.

Finally, Fig 9 in S1 Text shows often observed case of a seizure where a large majority of the regions is inferred to be recruited in the seizure at approximately same time. None of the regions is strongly inferred as epileptogenic, since it is difficult to identify which regions are the leaders and which are the followers of the seizure activity when the temporal delays of the recruitment are this small.

## 2.6 Leave-One-Out cross-validation

The goal of the Leave-One-Out (LOO) cross-validation is to assess whether we can correctly predict the seizure state and the onset times of the hidden regions. By definition, the

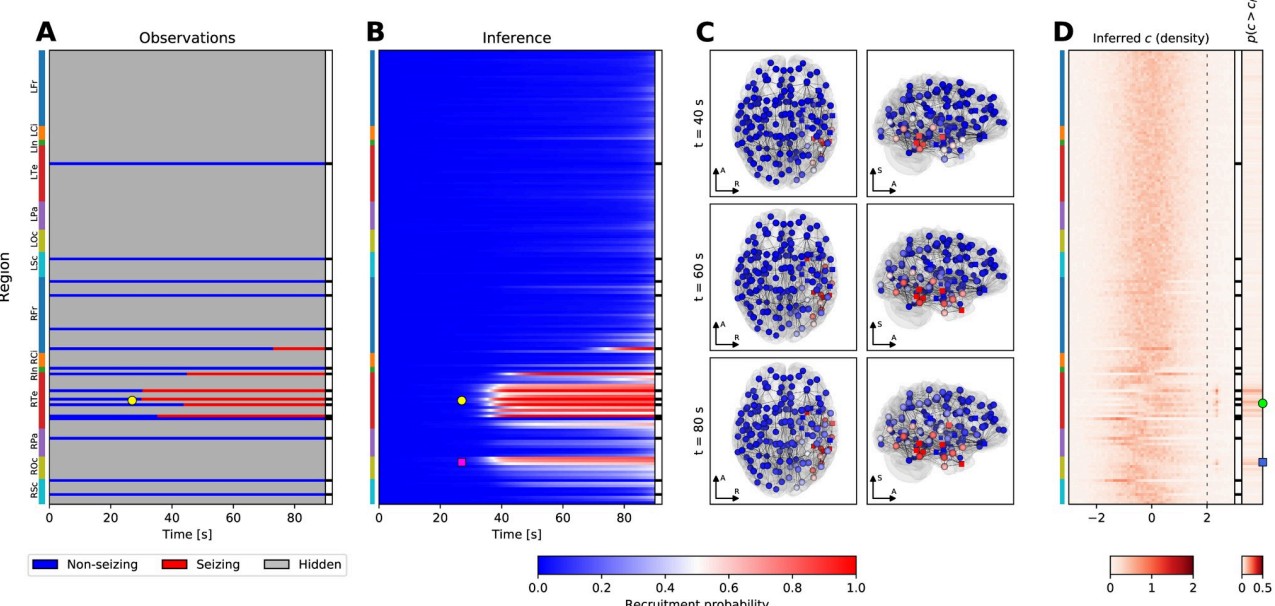

**Fig 5. Example of the inference results on a seizure from subject 33.** (A) Observation of the seizure. On vertical axis are the brain regions with color bars on the left indicating the anatomical grouping (abbreviations as in Fig 2). On the horizontal axis is the time; the onset time of the first observed seizing region is always aligned to $t = 30s$. The black and white column on the right shows which regions were observed. (B) Results of the inference. The partial observations from panel A are completed by the inference; the plot shows the recruitment probability $r_i(t) = p(t_i \leq t)$, i.e. the posterior probability that a region $i$ is recruited at time $t$. In the observed regions the inferred probabilities follow closely the observations with some blurring around the onset due to the assumed observation noise. (C) Snapshots of the recruitment probabilities at three time points. Color code same as in panel B. The spheres and the cubes represent the hidden and observed regions respectively. Only top three percent of the strongest connections are shown for visual clarity, their thickness is proportional to the maximum of the two oriented connection strengths. Axes notation: R, Left-Right axis; A, Posterior-Anterior axis; S, Inferior-Superior axis. (D) Inferred excitability. Left subpanel shows the posterior distributions of the excitabilities, dashed line indicates the threshold of high excitability at $c_h = 2$. Right subpanel shows the probability of high excitability $p(c > c_h)$. (A-D) The seizure is observed to start and remain restricted in the right temporal lobe (yellow circle, panels A, B). The result of the inference in addition points to the involvement of several regions in the right occipital lobe (magenta square, panel B). The regions identified by the inference as possibly epileptogenic are mainly located in the right temporal lobe (green circle, panel D), however, the inference also points to a possible epileptogenic zone in the right occipital lobe (blue square, panel D).

information about the hidden nodes is not available, thus we cannot evaluate this directly. Instead, we adopt the LOO approach: For every seizure and every observed region we fit the single-seizure model to the data with the observation of that specific region left out. Then we can compare how well the prediction matches the left-out observation. We note that the LOO analysis is done on the region (and not the channel) level. Given that multiple contacts of a single electrode (or several electrodes) are often assigned to a single region, this approach is loosely comparable to leaving out multiple channels at once, with the benefit of allowing us to focus on the role of the structural network (and not the source-to-sensor projection) in the prediction.

To avoid reusing the same data twice—once for the fitting of the multi-seizure model and once for the LOO fitting—all seizures from one fold are fitted using the hyperparameters obtained from the other fold. In addition to the LOO fitting, each seizure was fitted also with no data excluded for the analysis in the next section. In total, 3027 inferences of the single-seizure model were run, each with two MCMC chains. From these, in 179 cases one of the chains was stuck and in 28 both chains were stuck; the latter cases were excluded from further analysis. In the remaining results, 99.80% of regional excitability parameters had values of split chain reduction factors $\hat{R}$ below 1.1 and effective sample size $N_{eff}$ above 30, indicating acceptable convergence for vast majority of the parameters.

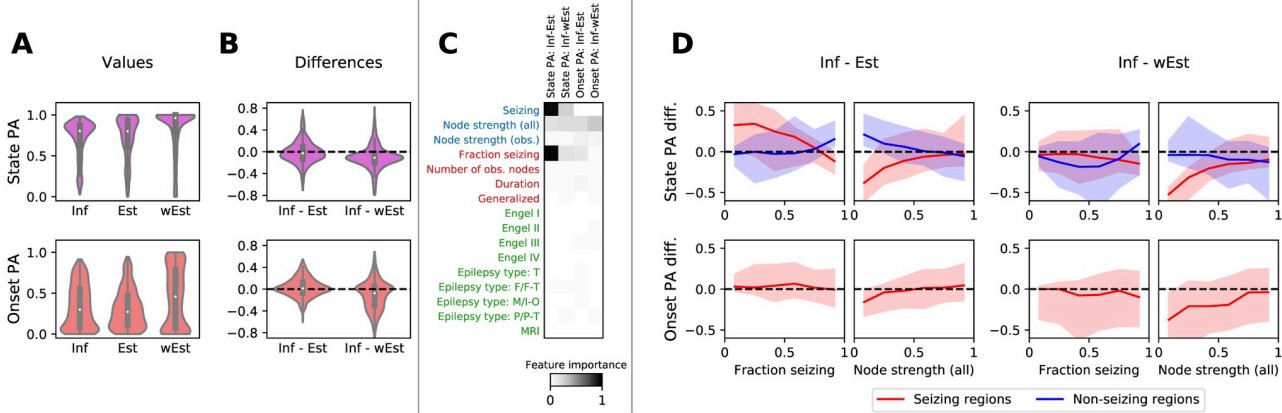

**Fig 6. Results of the leave-one-out (LOO) validation.** (A) Distributions of the state prediction accuracies (PA), top, and onset prediction accuracies, bottom, calculated with the inference (Inf) method and the unweighted estimate (Est) and weighted estimate (wEst). In the upper panel, each data point ($n = 2863$) corresponds to one observed region in one seizure ($n_{seizures} = 137$) of one patient ($n_{patients} = 44$) for which LOO analysis was successfully performed. In the lower panel, only the seizing regions are considered ($n = 1679$). (B) Pairwise differences between the accuracies obtained by the inference and the two estimates. In both A and B, the violin plots show a kernel density estimate of a given variable. The inner boxplots show the median (white dot), interquartile range (IQR, gray bar) and adjacent values (upper/lower quartile +/- 1.5 IQR, gray line). (C) Feature importances for predicting the differences in B. On vertical axis, features on region level (blue), seizure level (red), and patient level (green) are shown. Horizontal axis contains the target variables, that is the four prediction accuracies differences. Higher values of feature importance indicate a stronger dependency of the target variable on the feature. Three features are identified as most relevant: seizing/non-seizing state of a region, node strength in a network, and a fraction of seizing region in a seizure. (D) The partial dependency plots of the PA differences on features identified by the feature importance analysis in C. These are the fraction of seizing nodes, and the node strength, both for seizing and non-seizing regions. Full line shows the median, and filled area represents the 10 to 90 percentile range.

The predictive power of the method was evaluated using the state prediction accuracy, quantifying the ability to predict the seizing or non-seizing state of a hidden region, and the onset prediction accuracy, quantifying the ability to correctly predict the exact onset time of a hidden region. To provide a baseline, the inference method was compared with two simpler approaches: an estimation based only on the onset times in other regions, and a weighted estimation based on the onset times in other regions and on the network structure. The results demonstrate that the seizing/non-seizing state of a region can be predicted with a median state prediction accuracy 80.5%, and the onset time can be predicted with median onset prediction accuracy 29.9% (Fig 6A). On group level, these values are however not better than those of both the unweighted and the weighted estimates (Fig 6B). It is worth noting that the distributions of prediction accuracies are constructed from all observed regions in all available seizures, and some patients have more recorded seizures than others. The distribution thus do not show any "patient-averaged" prediction accuracy, rather the performance on the whole data set. Nevertheless, the following results indicate that the patient-specific effects that might bias the distribution are weak.

To analyze for which regions, seizures, or patients the method performs better (or worse), we carried out a feature importance analysis (Fig 6C). We fitted a gradient boosting regressor, using the prediction accuracies as the target variables, and as predictors we used multiple region-level, seizure-level, and patient-level variables. Then we calculated the permutation feature importance, which quantifies how much the prediction performance drops if the feature data is randomly permuted. The results highlight three features as important: seizing/non-seizing status of a region (region-level), node strength of a region (region-level), and a fraction of seizing regions in a recorded seizure (seizure-level). Notably, other seizure-level features (number of observed nodes, seizure duration, or whether a seizure is secondarily generalized) nor patient-level features (Engel score, epilepsy type, and MRI lesion findings) do not have a

strong effect on the accuracies. Qualitatively similar results were obtained also when different regression methods were used (Fig 10 in S1 Text).

The dependency on these three features is analyzed in Fig 6D. Two remarks can be made. First, the results show a robust improvement obtained by the inference over the unweighted estimate for the seizing regions in seizures where the majority of the observed regions do not seize, especially for the state prediction accuracy. In other words, the inference is better than the unweighted estimate in finding the hidden seizing regions in seizures where majority of the observed regions are not recruited, such as the seizure depicted on Fig 5. This improvement is however not observed when compared to the weighted estimate.

Second, there is a drop in accuracy for weakly connected seizing nodes, present for both state and onset prediction accuracies compared both with the weighted and unweighted estimate. Such result can be understood considering that for an unconnected region in the absence of any information on excitability, the method will predict the region as non-seizing, as it cannot be pushed to seizure state via the network effects. This prediction would be wrong for seizing nodes. That is not a problem for the estimate methods; even the weighted estimate considers only the relative weights of its neighbors and not the absolute values.

Finally, we have analyzed the results on a subject and seizure level using a multi-level hierarchical model that accounts for the discovered dependency on the node strength and fraction of seizing regions in a seizure (S1 Text/Subject-level analysis of the prediction accuracy). Results are presented on Figs 13 and 14 in S1 Text. The analysis shows that individual variation in accuracies remains even after the accounting for the node strength and fraction of seizing regions, both at the subject and seizure level. Detailed analysis of the variations is out of scope of this work, nevertheless, these results can guide the future investigation of the model strengths and weaknesses on the subject and seizure level.

## 2.7 Validation against the resected regions

In addition to the predictions of the onset times in hidden regions, the product of the inference is also a spatial map of the excitability parameter $c$ for every seizure. We evaluated whether this excitability is useful for localizing the epileptogenic zone. No ground truth to directly validate against exists, therefore we employed the following methodology: we restricted ourselves only to the patients that were operated and for which the post-operative MRI was available ($n = 18$). We extracted which brain regions were resected, and using the inferred excitabilities for a specific patient and a specific seizure we performed an *in silico* resection, that is, we removed the resected regions from the network model and we performed the forward simulations with the excitabilities inferred from the observed seizures (Fig 7A–7C). If the fitted model represents the reality well, one can expect that the successful surgeries that stopped the seizure occurrences in the patient will stop the seizures also in the computational model, and that the failed surgeries that did not reduce the seizure occurrences will fail in the computational model too. In other words, real-world outcome classified by the Engel score should correlate with *in silico* outcome.

The virtual resection was rarely sufficient to stop the seizures, indeed, large amount of brain regions continued seizing after the virtual surgery in patients of all Engel classes (Fig 7D, top panel). However, the resection was more successful in relative reduction of the number seizing regions among the patients with Engel score I and II than among Engel III and IV patients (Mann-Whitney $U = 67$, $n_{I,II} = 10$, $n_{III,IV} = 8$, $p = 0.019$; Fig 7D, bottom panel). Here by relative reduction we mean the number of regions where seizure activity was suppressed over the number of initially seizing regions. Similar results were obtained also when evaluating the overall

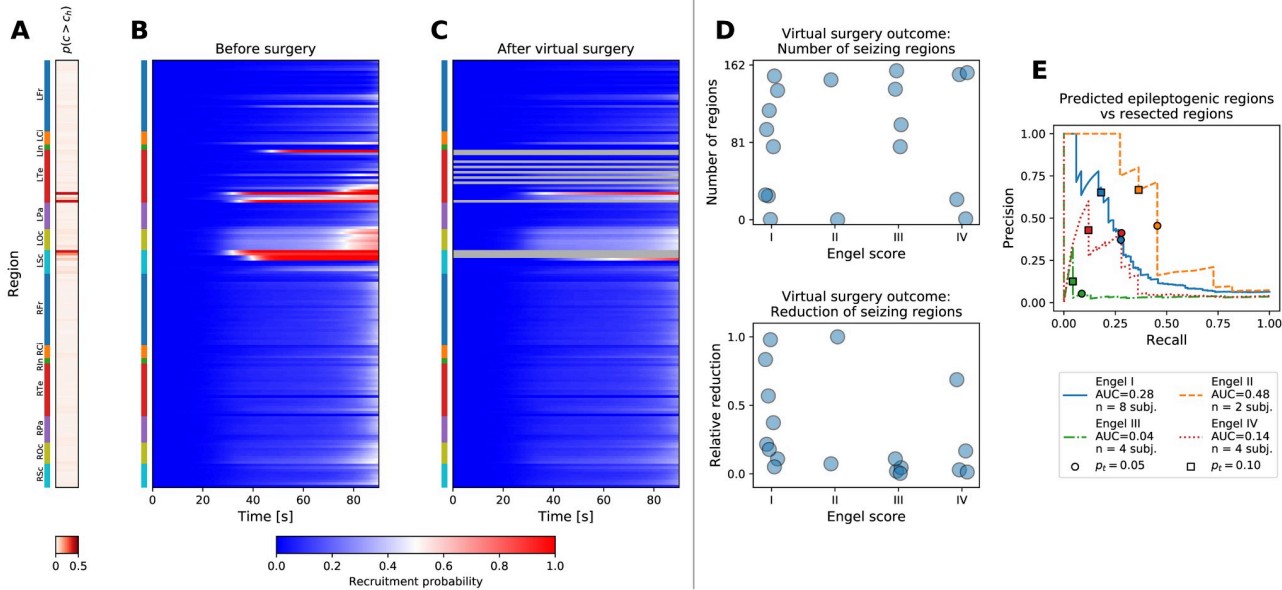

**Fig 7. Virtual resection.** (A-C) Example of a virtual resection on a seizure from patient 17, cf. Fig 7 in S1 Text. (A) Inferred probability of high excitability. Anatomical abbreviations as in Fig 2. (B) Pre-operative seizure dynamics as inferred from the data. (C) Post-operative seizure dynamics. The resected regions were removed from the model, and the dynamics was simulated using the excitabilities inferred from the pre-operative observations. The seizure activity is not completely stopped, but the number of seizing regions is reduced. (D) Outcome of virtual surgeries on a group level. Each point corresponds to an operated patient ($n = 18$). Top panel shows the number of post-operative seizing regions $n_{postop}$ (i.e. the regions with recruitment probability above 50% at $t = t_{lim}$), bottom panel shows the relative reduction of the seizing regions compared to the pre-operative level, ($n_{preop} - n_{postop})/n_{preop}$. For patients where multiple seizures were available, the values were averaged across seizures. (E) Precision-recall curves for evaluating the match between the performed resection and the inferred epileptogenicity. The precision and recall values were calculated for varying threshold $p_t$ on high epileptogenicity, $p(c > c_h) > p_t$; the threshold on high excitability $c_h = 2$ was kept constant.

decrease in seizure probability, which avoids the thresholding to determine if a region is seizing or not, and when taking both the average or minimal value of the relative reduction across seizures for patient when multiple seizures were available (Fig 11 in S1 Text). Neither the real nor the virtual surgery outcome shows a strong dependence on the number of resected regions in the examined patient data set (Fig 12 in S1 Text).

Further insight can be gained by a direct analysis of the excitability results without relying on the virtual resection. We have evaluated the match between the regions that were identified as epileptogenic by the model and the regions that were surgically resected (Fig 7E). If the model is to be useful, we expect that these should overlap more in the subjects where surgery was successful, and less where the surgery failed. To analyze the match, we plot the precision-recall curves. Precision-recall analysis is suitable for binary classification tasks on unbalanced data sets where number of true negatives (that is, non-resected regions not identified as epileptogenic) outweights the number of positives [38]. In this context, the precision expresses how many of the predicted epileptogenic regions were resected, and the recall expresses how many of the resected regions were predicted to be epileptogenic. These values are computed for varying threshold on what constitutes a highly epileptogenic region. A perfect match would result in a curve passing through top-right corner, while bottom-left corner indicates a mismatch. In the results we indeed see better performance among the Engel I and Engel II classes than Engel III and Engel IV classes, even if the performance does not follow the surgery outcome precisely: our results show the order of Engel classes II > I > IV > III.

## 3 Discussion

### 3.1 Main results

In this work we have developed and evaluated a novel method to infer the seizure propagation patterns in partially observed brain networks. The method is transparent in its assumptions and it is data-driven, meaning that the model hyperparameters are inferred from the patient cohort data, and only few clearly interpretable parameters need to be set by hand. It performs reliably when applied to synthetic data generated by the same model, although with limits caused by the incomplete observations of the seizures (Fig 3 in S1 Text). When predicting the states and onset times of the hidden regions in real data, the method performs better than the empirical estimate that does not use the network structure, but not better than the weighted estimate using the network structure (Fig 6). While this result may be initially viewed with disappointment, it is important to keep in mind that unlike the weighted estimate our method provides a generative model which can be interrogated and its assumption can be questioned and modified. It is thus amenable to further development which may lead to more precise predictions. Furthermore, it provides the estimate of regional excitabilities. Indeed, the comparison with surgically resected regions and the surgery outcome indicates a link between the inferred excitabilities and the epileptogenic zone in the analyzed patients (Fig 7).

### 3.2 Role of the large-scale structural connections in seizure propagation

Focal seizures are known to spread both locally and distally, but while recent experiments using *in vivo* rodent models shed more light on the mechanism of the long-range propagation [39, 40], it's role in the large-scale spatio-temporal organization of epileptic seizures in human patients is not yet sufficiently understood. Here we have shown that a method based on the principles of network propagation predicts the states of the hidden regions better than an empirical estimate that does not use the network structure, but not better than an empirical estimate utilizing the network structure. Such results confirm the core tenet of our method that seizures spread along the long-range structural connections [11], possibly supported by the alteration of whole-brain structural networks observed in epileptic patients [8]. Our results further indicate that the network influence is strong enough not only to predict where a seizure will spread from known origin, but also in some cases identify this epileptogenic zone even if it is not directly discovered by the implantation only due to the effects it has on the connected observed regions (Fig 4 in S1 Text).

### 3.3 Reliance on the incomplete observations

Given that the method is trying to fill in incomplete data, it should be no surprise that there are limits of what can be inferred. Estimation of the onset times of hidden regions are only probabilistic and sometimes inaccurate, and the hidden epileptogenic zones can be discovered only when they are well connected to the observed regions. The straightforward way to reduce the uncertainty in the inference results is to observe more network nodes, however, the number of implanted electrodes is limited due to the clinical considerations. Other ways may be possible though. Even when the number of the observed nodes is fixed, it is possible that the uncertainty could be reduced by careful selection of the observed nodes before the implantation. The question of the system observability and the choice of the optimal sensor placement in complex networks is a problem well investigated in the field of control theory [41, 42]. However, most results pertain to linear systems or nonlinear systems with polynomial or rational coefficients, and are thus not applicable to our system based on the threshold dynamics,

especially considering that the trajectory through the high-dimensional state space cannot be *a priori* estimated. Furthermore, efficient analysis of large networks still poses a research problem.

Other way to increase the number of observed nodes would be to use more advanced technique to invert the SEEG signals to the space of brain regions, compared to here employed nearest-region mapping. Studies have demonstrated that source localization techniques may be beneficial not only in their traditional applications with non-invasive EEG and MEG recordings, but also with invasive recordings for interictal spike localization [43, 44]. Extending such approach to ictal recordings could lead to better exploitation of the data and thus to more precise and extensive seizure observations on the level of brain regions.

### 3.4 Assumptions of the method

The method introduced here rests on several assumptions regarding the nature of seizure dynamics. The guiding principle during the method development was simplicity over complexity with possibly irrelevant details; that was in order to build and analyze a transparent and robust method on which further, more refined versions can be based in the future. Some of the assumptions taken can be however questioned and we discuss them here.

**Binary healthy/seizure state.**   Our method rests on the notion that the state of a brain region during a seizure can be clearly separated to either healthy or seizure state. That is an assumption shared with other network models of seizure propagation; variations on a dynamical system with a fixed point and a limit cycle representing the healthy and the seizure state respectively are commonly used in the network models [15, 18, 19]. It is a useful conceptual simplification, but it a simplification nevertheless. While the electrographic recordings in human patients are commonly divided into interictal, preictal, ictal, and postictal periods, their boundaries or even their existence may not always be clear [45]. Multiple different electrographic patterns are observed in intracranial recordings at seizure onset [46–48], which likely reflect different underlying dynamics of the neuronal tissue during seizure initiation and spread [36, 49]. Recent experimental studies paint a complex picture of spreading seizures with ictal core organized by fast traveling waves, ictal wavefront, and surrounding areas affected by strong feedforward inhibition [40, 50–52]. All of these can be considered abnormal states, yet all are distinct with different roles in seizure spread and with different contributions to the electrographic signals. As it is, the present inference method relies on the model simplicity obtained by having only two states. Nevertheless, introducing more complex dynamics while retaining the conceptual simplicity can be envisioned and should be pursued as the role of the different dynamical regimes in seizure organization and spread as well as their relation to the observable signals becomes clearer.

**Normal distribution of the excitabilities.**   We set the prior distribution of the regional excitabilities to a normal distribution. This choice can be justified from two perspectives. First is that of information theory: the normal distribution has maximum entropy among the distributions with known mean and variance [53] and thus in this sense it is the weakest assumption that can be made. Second, the regional excitability is an abstraction representing the cumulative effect of the underlying components that play a role in the ictogenesis. By virtue of the central limit theorem [54], if these are independent random variables, then their sum converges to the normal distribution, no matter what their original distribution is. It is however worth considering that localized structural abnormalities such as focal cortical dysplasias or brain tumors are among the common causes of epilepsies [55, 56]. While in such cases a clear distinction between the healthy and the affected tissue exists and a bimodal prior distribution might seem more appropriate, quantitative analysis of the intracranial signals

indicates that also the structures outside of the lesion might have elevated epileptogenicity [57, 58], therefore, the relation between the structural abnormalities and epileptogenicity is not as direct.

**Unbiased implantation.** The inference method introduced here implicitly assumes an unbiased implantation of the intracranial electrodes, i.e. that the observed regions are selected randomly. That is decidedly untrue, as the electrodes are implanted in the regions suspected of the seizure involvement based on the existing non-invasive data [5, 59]. Presumably it is this assumption that leads to the inferred widespread recruitment patterns such as shown on Fig 9 in S1 Text. Technically this bias can be implemented in the model by adding an appropriate constraint into the statistical model, however two problems remain. First, the implantation might not always perfectly correspond to the clinical hypothesis of involvement; some suspected regions might not be implanted due to the practical constraints of the electrode implantation such as the avoidance of blood vessels, safe distance between electrodes, or suitable entry angle through the skull. Second, and more fundamental, is the issue of validation. Any prior placed specifically on the hidden regions will primarily affect those hidden regions, and its effect on the observed regions will be only secondary and presumably minor. However, in the utilized leave-one-out framework, only the observed regions can be left out and thus only the predictions for the observed regions can be validated. In other words, current framework is not sufficient to properly evaluate the effects of the biased implantation assumption, and other approach would be needed.

**Shared seizure dynamics.** Focal seizures exhibit considerable variability between patients in their causes [55, 56], electrographic onset patterns [46, 48], their duration [60], or underlying dynamic [32]. Even in individual patients, while more stereotypical [61–63], the seizures may still differ markedly in the extent of the recruitment or in the seizure duration [52, 60, 64], particularly considering that our data set contains both focal and secondarily generalized seizures. In our model we instead allow this variability to be explained by the seizure-specific excitabilities in conjunction with the patient-specific structural connectomes. It is possible that such approach is not sufficient to fully capture the variability of seizure dynamics, and that varying the excitation function $f_q$ across the seizures and patients would lead to better results. Multiple options to do so are at hand, including the inference of the parameterization $q$ on a single seizure or single patient level, or inferring the parameterization for specific seizure classes, either predefined or extracted from the data via appropriate unsupervised learning method.

### 3.5 Towards non-invasive EEG?

In the present work we use intracranial EEG to inform us about the seizure evolution. Given the medical risk of the invasive method for the patient [3], the appeal of exploiting non-invasive EEG recordings instead is obvious. Our model operates in the source space, that is on the level of brain regions, and thus a sensor-to-source projection is needed prior to running the method. With intracranial EEG we employed a straightforward distance-based mapping. The source inversion of scalp EEG however poses much larger challenge despite the advances in EEG source localization techniques and increased availability of high-density EEG [65]. In the field of epilepsy the focus remains on localizing either the interictal spikes, or, in case of ictal source inversion, on localizing the seizure onset zone using the recording from the seizure onset only [66]. Mapping the entire seizure propagation to source space, as our method would require, is a problem not sufficiently tackled yet. Exploiting the non-invasive data in the current framework thus awaits the development and validation of robust source inversion method for spatially extended and time-varying sources.

### 3.6 Conclusions

Wide range of models of seizure dynamics exist today, ranging from detailed single-neuron models to network-based whole brain models [67], with more recent studies attempting to link the models to patient-specific intracranial recordings via parameter inference on regional scale [30, 32] or at a whole-brain scale [33]. In this work we introduce a method of the latter type. Here the seizure dynamics is extremely simplified, however, we do not discard the considerable complexity of seizure dynamics on micro- and meso-scale levels. We rather explore how much of the observed spatio-temporal organization can be explained by the simple principles encoded in our model that hides this complexity behind a single regional excitability parameter. In exchange for this simplification we obtain a model that can be reliably inverted and can exploit the intracranial electrographic data on the whole-brain scale not only for a single patient, but also for patient cohorts. The epilepsy models at different scales are mutually complementary, and it is by bridging the gap between the different levels that the patient-specific seizure dynamics on the whole-brain scale can be understood. We expect and hope that as the understanding of epileptic seizures on smaller scales progresses, its incorporation in the whole-brain models such as the one presented here will lead to the desired goal.

## 4 Materials and methods

### 4.1 Ethics statement

The approval was granted by the local ethics comittee (Comité de Protection des Personnes Sud-Méditerranée I); the patients signed a written informed consent form according to its rules.

### 4.2 Patients and data acquisition

Fifty epileptic patients who underwent standard clinical evaluation for surgery candidates at La Timone hospital in Marseille were selected for the inclusion in this study. Details of the subjects are given in Table 3 in S1 Text. The evaluation included non-invasive T1-weighted imaging (MPRAGE sequence, either with repetition time = 1.9 s and echo time = 2.19 ms or repetition time = 2.3 s and echo time = 2.98 ms, voxel size 1.0 x 1.0 x 1.0 mm) and diffusion MRI images (DTI-MR sequence, either with angular gradient set of 64 directions, repetition time = 10.7 s, echo time = 95 ms, voxel size 1.95 x 1.95 x 2.0 mm, b-weighting of 1000 s mm$^{-2}$, or with angular gradient set of 200 directions, repetition time = 3 s, echo time = 88 ms, voxel size 2.0 x 2.0 x 2.0 mm, b-weighting of 1800 s mm$^{-2}$). The images were acquired on a Siemens Magnetom Verio 3T MR-scanner.

The invasive evaluation consisted of implantation of multiple depth electrodes, each containing 10 to 15 contacts 2 mm long and separated by 1.5 or 5 mm gaps. The SEEG signals were recorded by a 128 channel Deltamed system using at least 256 Hz sampling rate. The recordings were band-pass filtered between 0.16 and 97 Hz by a hardware filter. Only the seizures with duration longer than 30 seconds were included in the analysis (see below for the reason why), and subclinical seizures were not considered. Apart from that, no other criteria for exclusion were applied. Five patients in the cohort did not have any seizures satisfying the duration criterion. Their structural connectomes, described in the following section, were constructed nevertheless, and they were used for generating and analyzing synthetic seizures (S1 Text/Validation on synthetic data). Exclusion of the short seizures left 45 patients for which at least one seizure was available.

After the electrode implantation, a CT or T1-weighted scan of the patient's brain was acquired to obtain the location of the implanted electrodes.

## 4.3 Structural model of the brain

The structural connectome was built with a reconstruction pipeline using generally available neuroimaging software. The current version of the pipeline evolved from a previously described version [68].

First, the command *recon-all* from FreeSurfer package [69] in version v6.0.0 was used to reconstruct and parcellate the brain anatomy from T1-weighted images. For reasons of conformity with established implantation and interpretation practices at the epileptology department of AP-HM, we employed a custom brain parcellation in this study [70]. This custom parcellation is based on the default FreeSurfer segmentation of the brain tissue [71] with the Destrieux cortical parcellation [72], where some regions are merged together, some are split into multiple parts, and yet others are split and then merged with existing ones. The full list of the performed operations is given in Table 1 in S1 Text. The resulting parcellation contains 162 regions with 72 cortical and 9 subcortical regions per each hemisphere (Table 2 in S1 Text). Due to the splitting of the largest cortical regions from the Destrieux atlas, the resulting parcellation contains cortical regions with sizes on the same order of magnitude [70]. While the method can be applied to parcels of uneven size, vast differences in sizes could lead at the same time to high computational costs (spent on the small regions) and low precision (limited by the large regions).

Next, the T1-weighted images were coregistered with the diffusion weighted images by the linear registration tool *flirt* [73] from FSL package in version 6.0 using the mutual information cost function with 12 degrees of freedom. The MRtrix package in version 0.3.15 was then used for the tractography. The fibre orientation distributions were estimated from DWI using spherical deconvolution [74] by the *dwi2fod* tool with the response function estimated by the *dwi2response* tool using the *tournier* algorithm [75]. Next we used the *tckgen* tool, employing the probabilistic tractography algorithm iFOD2 [76], to generate 15 millions fiber tracts. The connectome matrix was then built by the *tck2connectome* tool.

Each element $\hat{w}_{ij}$ of the generated connectome matrix $\hat{W}$ represents the number of fibers from region $j$ to region $i$; note that we did not use the log-count of the fibers. The elements of the matrix were then scaled by the volumes of the target regions, $\tilde{w}_{ij} = \hat{w}_{ij}/V_i$, where $V_i$ is the volume of $i$-th region. This way, the elements $\tilde{w}_{ij}$ represent the density of the fibers projecting from region $j$ in region $i$. Afterwards, a correction for hippocampus connections was introduced: our custom brain atlas has the hippocampus split into two brain regions—anterior and posterior hippocampus. To correct for the presumably strong gray matter connections between the two parts, which are not discovered by the white matter tractography, the connectome matrix elements corresponding to the connection between the anterior and posterior hippocampus are increased by the value of 98 percentile of the connectome weights. Finally, the connectome is normalized so that the maximal sum of ingoing projections is equal to one: $w_{ij} = \tilde{w}_{ij}/\max_i(\sum_j \tilde{w}_{ij})$.

The electrodes contacts were localized using GARDEL software [77], available at https://meg.univ-amu.fr/wiki/GARDEL:presentation. Using the CT scan, the software automatically detects all SEEG electrodes and localizes the belonging contacts. These coordinates were then projected back to the reference frame of T1-weighted images using the transformation obtained by the linear registration tool *flirt*.

## 4.4 Resection masks and surgery outcome

For the patients who underwent resective surgery, the extent of surgical resection with respect to the anatomical parcellation according to the in-house custom atlas was defined using the

EpiTools software suite [77]. In brief, patient's specific 3D maps with SEEG electrodes were created and all the contacts were assigned to the anatomical regions of the atlas projected in the pre-operative MRI space. The co-registration of the post-implantation CT scan with electrodes with the post-operative MRI scan, and that of the post- and pre-operative MRI scans were then performed. The resected contacts were identified and assigned to the respective regions as defined by the pre-operative MRI. The regions resected but not explored by depth electrodes were then identified on the pre-operative/post-operative MRI co-registered images manually (SMV, JS). The completeness of resection of each respective region was estimated visually by two trained clinicians (FB, JS) and expressed in percentages. For the purpose of comparison with model predictions, only the regions with the resection extent above 50% (non-inclusive) were considered.

The surgery outcome was classified using the Engel score [78], where class I corresponds to patients free of disabling seizures, class II to patients with rare disabling seizures, class III to worthwhile improvement, and class IV to no worthwhile improvement. Postoperative seizure outcome was assessed at the latest available follow-up according to the Engel classification [78], i.e. at one year or later following surgery.

## 4.5 Onset time detection

The onset times were detected on the SEEG recordings in bipolar representation. For each bipolar channel, the onset time was detected by the following sequence of steps: First, the log-power in two frequency bands ($LP_{\text{low}}(t)$ from 1 to 12.4 Hz encompassing the $\theta$ and $\alpha$ bands; $LP_{\text{high}}(t)$ from 12.4 to 100 Hz encompassing the $\beta$ and $\gamma$ bands) was calculated by computing the time-frequency representation using the multitaper method, summing over the given frequency band. This division follows the use of the two bands in quantifying the epileptogenicity of the brain structures [4]. Next, the log-power time series were normalized to preictal baseline, determined as the mean log-power in the 60 seconds preceding the seizure onset as marked by a clinician:

$$LP_{\text{low,high}}^{\text{norm}}(t) = LP_{\text{low,high}}(t) - \langle LP_{\text{low,high}}^{\text{norm}} \rangle_{\text{preictal}}$$

Then the binary seizure mask was created:

$$\text{mask}(t) = [LP_{\text{low}}^{\text{norm}}(t) > \log(\delta) \quad \text{or} \quad LP_{\text{high}}^{\text{norm}}(t) > \log(\delta)]$$

where $[P]$ is the Iverson bracket,

$$[P] = \begin{cases} 1 & \text{if } P \text{ is true,} \\ 0 & \text{otherwise,} \end{cases} \tag{2}$$

and $\delta = 5$ is the prescribed threshold, corresponding to five-fold increase (or decrease) of power in a given band. Next, the mask was cleaned up to remove short intermittent intervals of seizure and healthy states. The mask was first smoothened by convolution with a rectangular window of duration 20 seconds and then binarized again with threshold 0.5. Finally, intervals of seizure state shorter than 20 seconds were removed. If there was no remaining seizure interval, then the channel was declared as non-seizing. Otherwise it was declared as seizing with the channel onset time equal to $t_{ch} = \min\{t \mid \text{mask}_{\text{clean}}(t) = 1\}$.

## 4.6 Mapping of channel observations to brain regions

The channel onset times were mapped onto the brain regions using the physical distance of the electrode contacts and the voxels belonging to the regions. For each bipolar channel, we

took the position of the midpoint between the two electrode contacts, and we computed the distance to the brain regions. If the two nearest regions were in a similar distance from the midpoint (i.e. satisfying $d_2/(d_1 + 0.5mm) < 2$, where $d_1$ and $d_2$ are the distances of the nearest and second nearest region respectively), we did not assign the observation to any region, otherwise we assigned it to the closest region. Note that we did not set any upper limit on the distance, and thus, in theory, contacts arbitrarily deep in the white matter with little relevant signal could be assigned to a brain region. In our study 99.3% of the assigned contacts were less than 3mm away from the gray matter tissue and the furthest was 4.47mm away, so we do not consider the issue particularly problematic here, however more cautious approach should be employed in future studies.

After the assignment is done, none, one, or multiple observations may be assigned to a single region. If there was no observation, the region was considered hidden. If there was one, region observation was set to the assigned channel observation, and if there were multiple observations assigned, the region observation was set as the median of the assigned observations (with lower interpolation if there was an even number of observations). To facilitate the median calculation, the non-seizing regions were considered to be seizing at $t = \infty$.

If there were no seizing regions after the mapping, we excluded the seizure from further analysis. Otherwise, the onset times were shifted so that the first region onset time was at $t_1 = 30s$, and the regions with onset time greater than $t_{lim} = 90s$ were set as non-seizing (see the following section for the reasoning).

## 4.7 Dynamical model of seizure propagation

**The model.**   At the core of the method is the following model of seizure propagation in a weighted network. For a network with $n$ regions, the model reads

$$\dot{z}_i = f_q\left(c_i, \ \sum_{j=1}^{n} w_{ij}H(z_j - 1)\right) \qquad \text{for } i = 1, \ldots, n \tag{3}$$

Here $z_i$ is the slow variable of region $i$. Parameter $c_i$ is the node excitability, and $W = (w_{ij})$ is the connectivity matrix, normalized so that $\max_i \sum_j w_{ij} = 1$ as described in Sec. 4.3. The function $f_q : \mathbb{R} \times [0; 1] \to \mathbb{R}^+$ is the *excitation* function, parameterized by the parameter vector $q$. Due to the scaling of the connectome, the second argument is guaranteed to fall in the interval $[0, 1]$. We require that the function $f_q$ is positive and increasing w.r.t. its first parameter. The positivity requirement guarantees that the slow variable can only increase and thus all regions are pushed only closer to the seizure state. The increasingness in $c$ is the only requirement that breaks the symmetry in $c$, and it assures that the larger values of $c$ can be interpreted as more excitable. We furthermore require that $f_q(c, y)$ is onto $\mathbb{R}^+$ for any $y$, that is to assure the existence of a solution to the inverse problems (see S1 Text). Finally, $H$ is the Heaviside step function, representing the switch from the healthy to the seizure state when the slow variable crosses the threshold $z = 1$. The onset time of a region $i$ is defined as $t_i = \min\{t \mid z_i(t) \geq 1\}$. The system is completed by the initial conditions $z_i(0) = 0$. For a known vector of excitabilities $c$, parameter vector $q$, and a connectome matrix $W$, the model uniquely defines the vector of onset times $t$. We use the following shorthand for this mapping:

$$P_{W,q}(c) = t. \tag{4}$$

**Temporal scale of the modeled seizures.**   Because the function $f_q$ is positive, the model implies that every region will start to seize at some finite time. In reality, this is not the case,

and to account for that in the statistical model we introduce a time limit of a seizure $t_{\text{lim}}$, and consider all regions with onset time larger than the limit to be non-seizing. Setting this constant however poses a challenge, as it is influenced by two contradictory considerations. First, we want $t_{\text{lim}}$ to be high enough to avoid losing information, since any region with detected onset occurring after $t_{\text{lim}}$ is considered non-seizing, and the exact timing information is lost. At the same time we however want $t_{\text{lim}}$ to be also low: the model does not model seizure offset, and assumes that every recruited region stays seizing (and possibly pushing the connected regions to seizure state) until time $t_{\text{lim}}$. This assumption is broken for every region which terminates its seizure activity before $t_{\text{lim}}$, therefore low $t_{\text{lim}}$ is desirable as well.

No perfect solution exists to this conundrum, particularly considering that the constant is shared for all seizures in the data set in the present version of the model. We set $t_{\text{lim}} = 90$, so that the interval between the first detected onset time and the seizure limit ($t_{\text{lim}} - t_1 = 60$ s) approximates the median duration of all seizures in our data set (65.95 s). This choice expresses the time scale that we consider relevant for seizure propagation. Furthermore we excluded from our data set all seizures shorter than half of that (i.e. 30s) to avoid contaminating the data set with seizures where the assumption that every region seizes until $t_{\text{lim}}$ is most glaringly broken.

In the future, this issue could be partially alleviated by seizure-specific models with varying $t_{\text{lim}}$, we consider it in Discussion.

**Parameterization of the excitation function $f_q$.** In general, the excitation function $f_q(c, y)$ can be parameterized in any fashion, provided that it guarantees its positivity and its increasingness in the first argument. In this work we opted for a simple approach of a bilinear function followed by an exponentiation. This approach has the advantage that only few parameters are needed, and therefore there is little risk of overfitting to the training set. On the other hand, the function might lack the expressivity that more complex approaches would provide.

Our parameterization starts with a helper bilinear function $\hat{f}_q(c, y)$, described by four values $q_{aa}, q_{ab}, q_{ba}, q_{bb}$ in the interpolation points $[c_a, y_a]$, $[c_a, y_b]$, $[c_b, y_a]$, $[c_b, y_b]$ with $c_a = -1$, $c_b = 1$, $y_a = 0$, $y_b = 1$, as shown on Fig 8. The function reads

$$\hat{f}_q(c, y) = \frac{1}{(c_b - c_a)(y_b - y_a)} \left( \quad q_{aa}(c_b - c)(y_b - y) + q_{ba}(c - c_a)(y_b - y) \right.$$
$$\left. + \quad q_{ab}(c_b - c)(y - y_a) + q_{bb}(c - c_a)(y - y_a) \right).$$

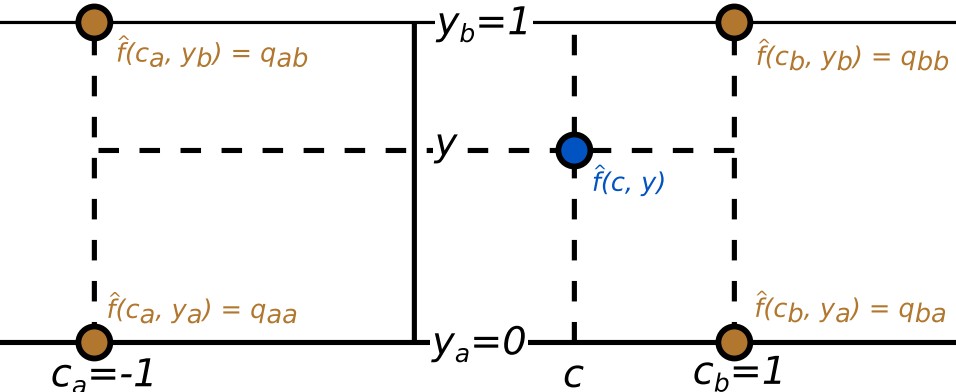

**Fig 8. Parameterization of the excitation function.** The helper bilinear function $\hat{f}_q(c, y)$ is described by four values $q_{aa}, q_{ab}, q_{ba}, q_{bb}$ in four interpolation points.

We then define the excitation function as $f_q(c, y) = \exp(\hat{f}_q(c, y))$, thus assuring that $f_q$ is positive. In order to assure that $f_q(c, y)$ increases in $c$, we declare $q_{ba} = q_{aa} + q_{ba}^*$ and $q_{bb} = q_{ab} + q_{bb}^*$, and we parameterize the function with the parameter vector $\boldsymbol{q} = (q_{aa}, q_{ab}, q_{ba}^*, q_{bb}^*)$ where $q_{aa}, q_{ab} \in \mathbb{R}$ and $q_{ba}^*, q_{bb}^* \in \mathbb{R}^+$.

## 4.8 Statistical model and the inference

Two statistical models are used in the study. The hierarchical model for multi-seizure inference (Box 1) is used for the learning of the hyperparameters. It takes the data from multiple seizures and infers the parameters $\boldsymbol{q}$ of the excitation function $f_q$. The low-level parameters—excitabilities $\boldsymbol{c}_k$ for all seizures $k$—are inferred as well, but due to the design of the leave-one-out validation scheme are not used in this study. The prior for the hyperparameters and its variance $\sigma_q$ specifically was selected as noninformative based on the observation that the parameters sampled from this prior distribution can produce wide range of behavior, including synchronized cascades of region onsets or nonsynchronous onsets in a model with no network effects.

The model for single-seizure inference (Box 2) is the simplification of the multi-seizure model, where the hyperparameters are fixed to given values, and the parameters of only one seizure are inferred. In this study, this model is used for the leave-one-out validation and the validation against the resection, where the hyperparameters are fixed to values learned by the multi-seizure model on training data.

The parameter inference with both multi- and single-seizure models was performed using the No-U-Turn Sampler [79], a self-tuning variant of Hamiltonian Monte Carlo method [80, 81] that eliminates the need to set the algorithm hyperparameters. The implementation in Stan software was used [82].

---

## Box 1. Statistical model for multi-seizure inference

**Input data**: Connectome matrices $W_k$, sets of `seizing` and sets of `nonseizing` nodes, onset times of seizing nodes $\tilde{\boldsymbol{t}}_{k,\text{seizing}}$.

**Parameters**: Hyperparameters $q_{aa}, q_{ab}, q_{ba}^*, q_{bb}^*$, excitability vectors $\boldsymbol{c}_k$.

**Constants**: $\sigma_q = 30$, $\sigma_t = 5$ s, $t_{\lim} = 90$ s

**Model**:

$$q_{aa}, q_{ab} \sim \text{Normal}(0, \sigma_q)$$
$$q_{ba}^*, q_{bb}^* \sim \text{HalfNormal}(0, \sigma_q)$$
$$\boldsymbol{q} = (q_{aa}, q_{ab}, q_{ba}^*, q_{bb}^*)$$

For $k = 1\ldots, n_{\text{seizures}}$:

$$\boldsymbol{c}_k \sim \text{Normal}(0, 1)$$
$$\boldsymbol{t}_k = P_{W_k, \boldsymbol{q}}(\boldsymbol{c}_k)$$
$$\tilde{\boldsymbol{t}}_{k,\text{seizing}} \sim \text{Normal}(\min(\boldsymbol{t}_{k,\text{seizing}}, t_{\lim}), \sigma_t)$$
$$t_{\lim} \sim \text{Normal}(\min(\boldsymbol{t}_{k,\text{nonseizing}}, t_{\lim}), \sigma_t)$$

---

---

### Box 2. Statistical model for single-seizure inference

**Input data**: Connectome matrix $W$, set of `seizing` and set of `nonseizing` nodes, onset times of seizing nodes $\tilde{t}_{\text{seizing}}$, parameters of the excitation function $q$.

**Parameters**: Excitability vector $c$.

**Constants**: $\sigma_t = 5$ s, $t_{\text{lim}} = 90$ s

**Model**:

$$c \sim \text{Normal}(0, 1)$$
$$t = P_{W,q}(c)$$
$$\tilde{t}_{\text{seizing}} \sim \text{Normal}(\min(t_{\text{seizing}}, t_{\text{lim}}), \sigma_t)$$
$$t_{\text{lim}} \sim \text{Normal}(\min(t_{\text{nonseizing}}, t_{\text{lim}}), \sigma_t)$$

---

For the multi-seizure inference, four independent MCMC chains were run each with random initiation in search space, and 500 steps in the warm-up phase and 500 generated samples. For the single-seizure inference, two chains were run, again with 500 steps in the warm-up phase and 500 generated samples.

## 4.9 Validation on real data

The leave-one-out cross-validation and the validation against the resection masks consisted of the following steps: First, we detected the onset times in all recorded seizures longer than 30 seconds, and excluded the seizures where no seizing region was detected. Next, the 44 subjects for which at least one seizure was available were divided in two folds (each with 22 subjects) and for both of these two folds the multi-seizure model was fitted independently of the other. We included at most two seizures from a single patient in the learning data set. We opted for two seizures as an imperfect compromise between the need for good generalization of the trained model (to which including large amount of possibly similar seizures from a single patient would be detrimental) and the wish to include as much data as possible. In the future, the optimal way to deal with uneven number of seizures would be to build a hierarchical model with subject and seizure levels that can account for that. If more than two seizures were available for one subject, the two seizures were selected randomly. This resulted in the inclusion of 21/18 patients with two seizures and 1/4 patients with one seizure in the first/second fold. In each fold the posterior distribution of the hyperparameters $q$ was obtained, and its mean was taken as a point estimate of the hyperparameters used in the subsequent steps.

With the inferred hyperparameters the leave-one-out validation was performed. For each fold, all seizures were repeatedly fitted with a single-seizure model, each time using the seizure data set with observation of one region excluded. The inferred states and onset times of the excluded regions were then compared to the left-out observations and evaluated using the measures described below. The hyperparameter values used for the single-seizure fitting in one fold were those obtained from the other fold, thus preventing reusing the same data for the model training and validation. To get the excitabilities used in the validation against the resection and surgery outcome, all seizures from both folds were fitted using the single-seizure model once more, this time with no observations left out. Again, the hyperparameter values

were those obtained from the other fold. The entire computational pipeline was constructed and run using the Snakemake workflow manager [83].

## 4.10 Evaluation measures

**State and onset prediction accuracy.** We define the *state prediction accuracy* for a region $i$ as the predictive probability that the region state is equal to the ground truth state conditioned on the data. We approximate this quantity using the posterior onset time samples, i.e.

$$P_i^{state} = \begin{cases} p(t_i < t_{\lim} \mid D) \approx \dfrac{1}{N}\sum_{s=1}^{N}\left[t_i^s < t_{\lim}\right] & \text{if region } i \text{ is seizing,} \\[2ex] p(t_i \geq t_{\lim} \mid D) \approx \dfrac{1}{N}\sum_{s=1}^{N}\left[t_i^s \geq t_{\lim}\right] & \text{if region } i \text{ is non-seizing,} \end{cases} \tag{5}$$

where $t_i$ is the onset time of region $i$, $D$ represents the onset times data (either from all observed regions for synthetic data validation, or with the region $i$ excluded for leave-one-out validation with empirical data), $N$ is the number of posterior samples, $\{t_i^s\}_{s=1}^{N}$ are the posterior samples of the onset time of region $i$, the constant $t_{\lim}$ is the artificial limit of a seizure, and $[\cdot]$ is the Iverson bracket (2). The motivation for this definition is the association of the regions that start to seize late (after time $t_{\lim}$) with non-seizing regions. The ground truth state is either the known simulated state of a hidden region in case of the synthetic data validation, or the state of a left-out observed region in case of the leave-one-out cross-validation on real data.

We define the *onset prediction accuracy* for a seizing region $i$ as the predictive probability that the onset time $t_i$ is sufficiently close to the ground truth conditioned on the data,

$$P_i^{onset} = p(|t_i - \tilde{t}_i| < T \mid D) \approx \frac{1}{N}\sum_{s=1}^{N}\left[|t_i^s - \tilde{t}_i| < T\right], \tag{6}$$

where $\tilde{t}_i$ is the ground truth, which is either the known simulated onset time in case of the synthetic data validation, or the left-out onset time in case of the leave-one-out validation on real data. The constant $T$ determines the temporal resolution of the measure, in this work we use $T = 5$ s. To avoid the border effect affecting the seizing regions with onset time close to $t_{\lim}$, we evaluate the onset prediction accuracy only for the seizing regions with $\tilde{t}_i < t_{\lim} - T$.

**Estimation methods.** To provide a point of comparison for the state and onset prediction accuracies obtained with the inference method, we introduce two simpler estimates of the region onset times. The calculation of the state and prediction accuracies in (5) and (6) for region $i$ uses the set of posterior samples of the onset time $\{t_i^s\}_{s=1}^{N}$. In the (unweighted) estimate we replace this set with the observed onset times of all other observed regions in the same seizure, $\{\tilde{t}_j\}_{j\in obs,\ j\neq i}$, where $obs$ denotes the set of observed regions. For the purpose of this computation we set the onset time of the observed non-seizing nodes to infinity. The state and onset predicted accuracies are thus calculated as

$$P_i^{state,Est} = \begin{cases} \dfrac{1}{n_{obs}-1}\sum_{j\in obs,\ j\neq i}\left[\tilde{t}_j < t_{\lim}\right] & \text{if region } i \text{ is seizing,} \\[2ex] \dfrac{1}{n_{obs}-1}\sum_{j\in obs,\ j\neq i}\left[\tilde{t}_j \geq t_{\lim}\right] & \text{if region } i \text{ is non-seizing,} \end{cases} \tag{7}$$

$$P_i^{onset,Est} = \frac{1}{n_{obs}-1}\sum_{j\in obs,\ j\neq i}\left[|\tilde{t}_j - \tilde{t}_i| < T\right], \tag{8}$$

In the weighted estimate we replace the set of posterior samples for region $i$ in (5) and (6) with a theoretical set of the onset times of the other observed regions $j$ in the same seizure, where every onset time is repeated proportionally to the weight of the connection between regions $i$ and $j$. In practice we do not build this set, we simply replace the averages in (7) and (8) with their weighted counterparts,

$$P_i^{state,wEst} = \begin{cases} \frac{1}{\sum_{j\in obs, \ j\neq i}(w_{ij}+w_{ji})}\sum_{j\in obs, \ j\neq i}(w_{ij}+w_{ji})\left[\tilde{t}_j < t_{\lim}\right] & \text{if region } i \text{ is seizing,} \\ \frac{1}{\sum_{j\in obs, \ j\neq i}(w_{ij}+w_{ji})}\sum_{j\in obs, \ j\neq i}(w_{ij}+w_{ji})\left[\tilde{t}_j \geq t_{\lim}\right] & \text{if region } i \text{ is non-seizing,} \end{cases} \tag{9}$$

$$P_i^{onset,wEst} = \frac{1}{\sum_{j\in obs, \ j\neq i}(w_{ij}+w_{ji})}\sum_{j\in obs, \ j\neq i}(w_{ij}+w_{ji})\left[|\tilde{t}_j - \tilde{t}_i| < T\right]. \tag{10}$$

**Feature importances.** To analyze the importance of various region-level, seizure-level, and patient-level factors on the method performance we carried out the feature importance analysis. Following predictors were considered: On region level,

- *Seizing*: Seizing/non-seizing status of a region,

- *Node strength (all)*: Node strength, that is $\sum_{j=1}^n(w_{ij}+w_{ji})$ for region $i$,

- *Node strength (obs.)*: Node strength towards the observed nodes, that is $\sum_{j\in obs}(w_{ij}+w_{ji})$ for region $i$;

  on seizure-level,

- *Fraction seizing*: Fraction of the seizing regions among the observed ones,

- *Number of observed nodes*,

- *Duration* of a seizure,

- *Generalized*: Is a seizure secondarily generalized or not;

  and on patient-level,

- *Engel*: Engel score, encoded in one-hot vectors,

- *Epilepsy type*: Four simplified epilepsy types. T (temporal), F/F-T (frontal / fronto-temporal), M/I-O (motor/insulo-opercular), P-PT (posterior / postero-temporal). Encoded in one-hot vectors.

- *MRI*: Normal or lesional MRI.

From these the variables (the differences inference- and estimate-based state and onset prediction accuracies) were predicted. The data were fitted with gradient boosting regression model [84] implemented in scikit-learn [85]. Then the permutation feature importance of feature $j$ was computed by reshuffling the feature $K = 30$ times, and for each repetition $k$ predicting the variable and calculating $s_{k,j}$ score ($R^2$ in our case). The feature importance is then defined as

$$i_j = s - \frac{1}{K}\sum_{k=1}^K s_{k,j}, \tag{11}$$

where $s$ is the score on uncorrupted data set.

**Precision and recall.** To compare how the predicted epileptogenicity correspond to the ground truth epileptogenicity (in case of synthetic data) or to resected regions (in case of real data) we use the measures of precision and recall. In this work, the predictive variable is the posterior probability that a region $i$ is highly excitable, $p(c_i > c_h) \approx \frac{1}{N} \sum_{s=1}^{N} \left[ c_i^s > c_h \right]$, where $c_h = 2$ is the threshold of high excitability. The value of $c_h = 2$ was chosen based on the visual inspection of the results as a value that sufficiently identifies the highly excitable regions (see examples in Fig 5 in the main text and Figs 7 and 8 in S1 Text). The regions with $c \geq 2$ are capable of autonomously switching to seizure state, that is, without any input from other recruited regions. As a reference, the threshold $c_h = 2$ corresponds to 3.69 epileptogenic regions out of 162 if the excitabilities follow the prior of standard normal distribution.

Given a threshold for the predictive variable we obtain a binary vector of predictions, which is compared with the binary vector of relevant elements. These are either the epileptogenic regions (for synthetic data) or resected regions (for real data). The comparison gives the number of true positives (TP; predicted and relevant regions), true negatives (TN; not predicted and not relevant regions), false positives (FP; predicted and not relevant regions), and false negatives (FN; not predicted and relevant regions). The *precision* is defined as $\frac{TP}{TP+FP}$ and *recall* as $\frac{TP}{TP+FN}$. These measures are well-suited for imbalanced data sets, where number of true negatives outweighs the other categories, as the number of true negatives does not enter in the calculation of the precision nor of the recall. Indeed, epileptogenic zone prediction is an example of such imbalanced data set, as the number of predicted and relevant regions is generally small relative to the number of brain regions.

## Supporting information

**S1 Text. Supplementary information for Data-driven method to infer the seizure propagation patterns in an epileptic brain from intracranial electroencephalography.**
(PDF)

## Acknowledgments

The authors wish to acknowledge the computational resources provided by the Swiss National Supercomputing Centre (CSCS) under project ID ich001.

## Author Contributions

**Conceptualization:** Viktor Sip, Viktor K. Jirsa.

**Data curation:** Huifang Wang, Julia Scholly, Samuel Medina Villalon.

**Funding acquisition:** Maxime Guye, Fabrice Bartolomei, Viktor K. Jirsa.

**Investigation:** Viktor Sip.

**Methodology:** Viktor Sip, Meysam Hashemi, Anirudh N. Vattikonda, Marmaduke M. Woodman.

**Resources:** Maxime Guye, Fabrice Bartolomei.

**Software:** Viktor Sip, Marmaduke M. Woodman, Huifang Wang.

**Supervision:** Viktor K. Jirsa.

**Visualization:** Viktor Sip.

**Writing – original draft:** Viktor Sip.

**Writing – review & editing:** Viktor Sip, Meysam Hashemi, Anirudh N. Vattikonda, Marmaduke M. Woodman, Huifang Wang, Julia Scholly, Samuel Medina Villalon, Maxime Guye, Fabrice Bartolomei, Viktor K. Jirsa.

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
