## [Decision Letter · Decision Letter 0]

30 Sep 2020

Dear Dr. Sip,

Thank you very much for submitting your manuscript "Data-driven method to infer the seizure propagation patterns in an epileptic brain from intracranial electroencephalography" for consideration at PLOS Computational Biology.

As with all papers reviewed by the journal, your manuscript was reviewed by members of the editorial board and by several independent reviewers.

The approach was overall appreciated. The significance of the data driven approach, not providing a clear novel biological insight, could be better appreciated if the results were more compelling, but of course data should not be twisted or tortured, and we appreciate your openness.

Speaking of which, the code should be made available by policy on a third party repository at the moment of acceptance, but I hope you agree to share it at the moment of resubmission at least with the editors and reviewers (thus it can be in a private repository or in an attached zip file).

In light of the reviews (below this email), we would like to invite the resubmission of a significantly-revised version that takes into account the reviewers' comments. In case of a revision which clarifies the technical issues, but still does not fully convince on the novel biological input, acceptance could be offered in PLOS One.

We cannot make any decision about publication until we have seen the revised manuscript and your response to the reviewers' comments. Your revised manuscript is also likely to be sent to reviewers for further evaluation.

Sincerely,

Daniele Marinazzo

Deputy Editor

PLOS Computational Biology

Daniele Marinazzo

Deputy Editor

PLOS Computational Biology

Reviewer's Responses to Questions

**Comments to the Authors:**

Reviewer #1: In this article, the authors develop a computational model of macroscopic brain activity. The model combines a patient’s structural connectome with observed brain activity from intracranial electrodes to infer brain electrical activity at unobserved brain regions during seizures. The presentation is clear, and the approach compelling. As the authors admit, the clinical results are mildly underwhelming (eg the model does not outperform empirical observations, the resection results are rather weak). But the framework is excellent. The model is simple and data-driven. The clear presentation of model assumptions is excellent. I believe this work is compelling and important to report to the scientific community. Please find below my comments, which I hope the authors find helpful.

Major Comments

1. Would the authors please provide the code used to (i) simulate an instance of the model, (ii) infer the hidden activity traces from the simulated data. For example, code to generate Figure 2 would be extremely helpful to enhance reproducibility of the work. Providing the code on a public repository (e.g., GitHub) and/or with the publication would be appreciated.

2. Do the results in Figure 3 depend on the method to detect the onset time? The method requires choosing a threshold and assigning a binary category to each channel. Would a different threshold impact the results?

3. The measures reported in Figure 4 seem quite complicated. Can the resection results be explored in another way, without performing a virtual surgery? For example, could the authors work directly with the patient clinical results, and compare excitability in a patient’s resected tissue vs in non-resected tissue, with the hypothesis that excitability is higher in resected tissue when Engel I or II? Or something similar? Maybe this analysis is done in Figure 7E, but that was difficult to understand.

4. The guiding principle of the method is simplicity over complexity (line 311). However, the clinical results of the model are weak. So, the argument could be made that the model is too simple to be useful. A stronger result in (3) would help counter this argument.

5. Could the authors comment on the applicability of this approach to other domains, eg, non-seizure activity, scalp EEG?

Minor Comments

Line 28-29. The two references refer to a simulation study and fMRI. Can the authors provide evidence for this claim for voltage activity? Or perhaps remove the work “strong” from the text.

Line 166-167, How to interpret R_hat and N_eff? Is N_eff = 30 good or bad? Can the authors provide 1 sentence definitions for each?

Line 182. Could the authors provide more explanation of Figure 4B in the main text? This Figure seems to be important, and the text in Results is maybe too concise.

Equation after line 523, Does the choice of function f matter? This choice seems rather complicated.

The authors might further motivate their model by citing the cellular automaton model of Traub in,

Traub, R. D., Duncan, R., Russell, A. J. C., Baldeweg, T., Tu, Y., Cunningham, M. O., & Whittington, M. A. (2010). Spatiotemporal patterns of electrocorticographic very fast oscillations (> 80 Hz) consistent with a network model based on electrical coupling between principal neurons. Epilepsia, 51(8), 1587–1597. http://doi.org/10.1111/j.1528-1167.2009.02420.

Reviewer #2: In this paper, the authors introduce a novel method to infer the ictogenicity of brain regions not mapped by intracranial EEG which may aid clinicians in the decision-making process preceding epilepsy surgery. The paper is well written, and the method is interesting and offers significant potential for further investigations and improvements.

Major issues

1. Despite the fact that the authors write that they considered 50 individuals, Table S3 seems to indicate that there were a number of them that did not have any recorded seizures, making them presumably not usable in this study. I suggest that the authors make this clear or remove the individuals from Table S3 while correcting the numbers mentioned throughout the text. It would also be useful to clarify how many individuals were considered in each analysis upon applying their criteria.

2. I found Fig. 6C and 6F and the accompanying text (lines 228-236) somewhat misleading, because the authors choose to present an analysis for the apparent sole purpose of boosting their method, while choosing to present only as a supplementary material (Fig. S9) a comparison that is not as supportive. I would suggest to either move panels 6C and 6F to Fig. S9, or merge S9 to Fig. 6. I also suggest editing the text in lines 228-236 accordingly.

Likewise, the statement in the abstract: "we demonstrate that the method can improve the predictions of the states of the unobserved regions compared to an empirical estimate" – this seems also somewhat misleading. While true, it conceals the fact that the method did not perform better than the ‘weighted estimate’.

(Nevertheless, I commend the authors for showing that a simple heuristic such as the ‘weighted estimate’ can produce results comparable to their method.)

Also, in lines 224-225, the authors wrote: "The results demonstrate high predictive power of the inference method". It is unclear whether it is high, because it seems comparable to the ‘baselines’ introduced. I suggest that the authors introduce perhaps another baseline such as a random predictor or apply their method to surrogates with randomized distributions of seizing regions and onset times.

3. I suggest that the authors add to Fig. 7 a ROC analysis (with AUC, sensitivity and specificity), as it is common in other studies of virtual resections.

Also, why did the authors choose to look at the reduction of the number of seizing regions, rather than looking at the reduction of the overall excitability (which has the advantage of avoiding an arbitrary threshold)? One could expect that successful surgeries should have a higher reduction of the relative brain excitability compared to unsuccessful surgeries. I suppose both are reasonable options, and thus both could be assessed (e.g. one could be shown as a supplementary figure).

Also, it may be questionable whether using the average values across seizures is the best option to present these results. For example, one could argue that the minimum reduction across seizures may be a better indicator, because if different seizures are driven by different mechanisms and hence providing different results, then one has to be as conservative as possible to make sure that all seizure generators are accounted for.

Minor issues

1. Introduction, line 62, the authors wrote: "In this work we approach the problem of seizure propagation in a network from the other side". It is not clear to me what "other side" means in this context.

2. Results, line 92: I suggest that the authors add here what w_ij is (for readers that didn’t read the Methods section).

3. Line 110: I suggest that the authors explain how the 90 sec were chosen and also explain why this arbitrary threshold should not make an impact on their analysis.

4. Lines 121-122, the authors wrote: "we want to infer the parameters q of the excitation function, which we assume are shared among all seizures". I suggest that the authors further clarify that they mean “all” the seizures from all the patients (and not all the seizures of each individual separately). The authors may also add that this is assumed for the sake of simplicity (and tractability?).

5. Lines 149-150, the authors wrote: "we observe that the state of over 90% of the observed regions is unambiguous". It is then clarified in the caption of Fig. 3 how the ambiguity is dealt with. I suggest that important details such as this that are stated in the figures’ captions should be either copied or moved to the main text (more examples below).

6. In the caption of Fig. 3, the authors wrote: " (H) Detected onset times of all seizing regions, relative to the clinically marked seizure onset." – I suggest that the authors further clarify that this is a time difference. The same applies to (I).

7. I suggest that the authors explain briefly what q_aa, q_ab, q_ba, and q_bb are in Fig. 4 (or in the accompanying text) for the convenience of readers that have not read the Methods section.

8. The number of effective samples is introduced in the paragraph in lines 165-170, but its significance and interpretation are not presented.

9. The threshold c_h=2 is introduced without explanations. I suggest that the authors explain its meaning, and how they chose this specific value. It may also be relevant to show how this threshold may impact on their results.

10. Concerning the results presented in Fig. 6, is the state prediction and onset prediction accuracies dependent on the location of the region being predicted? One could expect that regions on the boundary of the mapped regions by intracranial EEG may be more difficult to assess. If not, then this may add further support for the proposed method. I suggest that the authors address this possibility.

11. I suggest to the authors to further interpret and explain the results presented in Fig. 7E in the text (around lines 252-255).

12. Methods, lines 412-413, the authors wrote: "Only the seizures with duration longer than 30 seconds were included in the analysis." – I suggest that the authors add a justification as for why they neglected shorter seizures.

13. At the beginning of section 4.4, the authors define low and high frequencies and distinguish between the two based on frequencies being lower or higher than 12.4Hz. Why 12.4Hz specifically?

14. From my understanding, the equation for the LP^norm should take the absolute value of the RHS (at the beginning of page 18).

15. Line 488, the authors wrote: "…we did not assign the observation to any region" – I was surprised by this. Why not assign it to both regions? I suggest that the authors justify their choice.

16. I suggest to the authors to further motivate and explain the parameterization function (lines 519-526).

17. I commend the authors for their interesting Supplementary Material. I suggest to the authors to add more mentions to it in the main text, particularly to the results presented there (pages 30 to 36).

Reviewer #3: Sip et al. investigate if a simple computational model of spreading activity simulated on patient-specific structural (white matter) networks can explain observed patterns of seizure spread in different seizures across a cohort of epilepsy patients. Although the validation of the results may be negative, I still find this a very instructive study that I enjoyed reading, and that deserves to be published.

I have some general comments for the authors, followed by technical questions that I hope will help the authors strengthen their results and improve their paper.

General comments:

- Figures 3G & 4B: One striking observation in both figures is that a disproportionate amount of seizures have all regions seizing at some point.

o I suspect these are secondary generalised seizures? These seizures should perhaps be excluded as the mechanism of the generalisation is most likely thalamo-cortical, which you do not explicitly model. I suspect this is also the reason for the mismatch in simulated vs observed data in Fig 4B 1st panel? If you do not have that information on the seizures, I think it is at least worth a discussion point.

o The arbitrary cut-off of 90 seconds (to determine if a channel is seizing) is rather strange. Why can you not use the actual seizure duration as the cutoff in the seizure-specific models?

o Related to this, it may be worth showing in another figure/panel what Fig 4B would look like using seizure-specific models?

- Figure 6 validation results. Overall it is unclear to me if these distributions represent the overall distribution across all validation iterations in each seizure in each patient, or some aggregate across iterations/seizures/patients? I have several related comments for this aspect:

o I simply don’t understand where n=4542 and n=1679 come from. Is this related to lines 214 & 215? If so, please clarify in the text.

o Would patients with many seizures, or many electrodes not bias the distribution shown?

o It is well-known that patient-specific effects exist, so an analysis on the patient level would greatly help understanding the data. For example, are the validation accuracies particularly low for patients with worse surgical outcomes? Is it related to any other patient level feature you list in table S3? Do other aspects like implantation design influence the validation accuracies?

o Seizure-specific properties have also been highlighted recently, and perhaps it is worth investigating if the same patient can have vastly different validation accuracies for their seizures? Could this serve as a tool to select one specific seizure per patient for the subsequent analysis on surgical outcome (Fig 7)? Clinically, one would not necessarily expect all seizures in a patient to be informative for the surgical strategy.

o Leave-one-out validation may be a bit too “weak” in this analysis, as very often, a single channel can simply be predicted by its neighbouring channels. This is also possibly why the model inference does not outperform the weighted null-model. You may get stronger results when you leave out an entire electrode shank, which is also simulating a more realistic clinical scenario.

o I think the results are also slightly overstated/overgeneralised in lines 224 – 227. Perhaps being more specific would help. E.g. “The results demonstrate that we can predict if a channel seizes at any point in the seizure with a median accuracy of 80.5%.” Etc. Of course, if these results are driven by particular patients/seizures, please revise accordingly. I feel similarly about other parts of the writing (e.g. abstract).

- In Figure 7D and all the results relating to surgical outcome, have you considered the effect of the size of the resection, or the total number of nodes removed? There are also some indications in the literature that size of resection may be related to surgical outcome, have you tested this? Other authors create a null-model for each patient (often performed as virtual “random resections” of the same size as the actual resection) to tackle this problem. It is hard to appreciate/interpret the results in this part of the paper otherwise.

Technical questions:

- When were the surgical outcomes assessed?

- Two different imaging protocols were used. A quick report that the main results did not differ by protocol should be made.

- I did not understand the reasoning behind the assignment of the channels to brain regions. “If the two nearest regions were in a similar distance from the midpoint (i.e. satisfying d2/(d1 + 0.5mm) < 2, where d1 and d2 are the distances of the nearest and second nearest region respectively), we did not assign the observation to any region, otherwise we assigned it to the closest region”. In an example: you wouldn’t assign a channel with d2=2 and d1=1 (2/(1+0.5)=4/3<2) to any region, but you would assign a channel with d2=20, d1=9 (20/(9+0.5)~=2.1>2) to the nearest region that is in this case 9 mm away? I don’t think a simple ratio is the way forward here, was this a typo?

- "Only the seizures with duration longer than 30 seconds were included in the analysis. For some patients only these short seizures were recorded; their structural data were used in the study nevertheless."

o The reasoning behind excluding seizures of less than 30 s is unclear. Depending on the pathology and patient, 25 s long seizures are perfectly “valid” and informative. Especially If they are the only type of seizure in a patient.

o Can you please also clarify if you generally considered all seizures in a patient? If not, what were the selection criteria?

o Were subclinical seizures included?

o If I understand you correctly, you use the structural connectivity only for some patients. This is only for the analysis of the multi-seizure model? If so, please clarify here.

- In the first part of results, please clarify what q is exactly, as you then report on qaa, qab etc. without further explanation. Simply changing the order of Methods and Results may also be helpful. It may also help the reader if in the schematic Fig 1C you clarify the role of q and c.

- Suggest to clarify if the cortical parcellation used contains ROIs of roughly equal size and its implications.

- “At most two seizures per single patient were used for the fitting [of the multi-seizure model I presume] to avoid biasing the model towards the patients with more recorded seizures.”: How did you select the two seizures? For patients with only one seizure: is that not biasing against them?

- “In our model we explain this variability by the seizure-specific excitabilities, and, to a lesser degree, by patient-specific structural connectomes.” Can you clarify when you assessed the role of the patient-specific connectome? This is an interesting point!

- Related to the previous point: Line 437-439: dividing wij (fibre count) by target region volume may be problematic. Target region volume covaries with total brain volume. The target region may also show patient-specific patterns of atrophy. Therefore, these networks may not be comparable across subjects. Please also clarify if log fibre count was used.

- Section 4.9: please clarify N.

**Have all data underlying the figures and results presented in the manuscript been provided?**

Reviewer #1: **No: **Please provide the code used to (i) simulate an instance of the model, (ii) infer the hidden activity traces from the simulated data.

Reviewer #2: None

Reviewer #3: **No: **PLOS states that all data should be made available unless there are ethical/legal concerns. I don’t think there are ethical concerns regarding the anonymised data here, so all the anonymised data should be made available.

PLOS authors have the option to publish the peer review history of their article (what does this mean?). If published, this will include your full peer review and any attached files.

Reviewer #1: No

Reviewer #2: **Yes: **Marinho Antunes Lopes

Reviewer #3: **Yes: **Yujiang Wang
---

## [Decision Letter · Decision Letter 1]

8 Dec 2020

Dear Dr. Sip,

Thank you very much for submitting your manuscript "Data-driven method to infer the seizure propagation patterns in an epileptic brain from intracranial electroencephalography" for consideration at PLOS Computational Biology.

You satisfactorily addressed several issues, resulting in a much better work. Still, a few important issues, outlined below, remain unaddressed.

We cannot make any decision about publication until we have seen the revised manuscript and your response to the reviewers' comments. Your revised manuscript is also likely to be sent to reviewers for further evaluation.

Sincerely,

Daniele Marinazzo

Deputy Editor

PLOS Computational Biology

Daniele Marinazzo

Deputy Editor

PLOS Computational Biology

Reviewer's Responses to Questions

**Comments to the Authors:**

Reviewer #1: The authors have addressed my comments. Thank you.

Reviewer #2: The authors have answered all my queries adequately.

I would like to just point out a few minor corrections:

1) Section 2.5, 1st paragraph: “epileptognic” -> epileptogenic

2) Section 2.6, the authors wrote: "Nevertheless, the following results indicate no patient-specific effects that would bias the distribution are weak." – I find the sentence hard to read.

3) Section 2.7, penultimate paragraph: "worst result" – I suggest adding what “worst result” means.

4) Section 3.3: the authors wrote: "Given that the method is trying the fill in incomplete data," – I assume the authors meant to write “to fill” instead of “the fill”.

Reviewer #3: I thank the authors for addressing some of my comments, but a few points remain open.

I don’t typically like knit picking if the details don’t have a great impact on the overall conclusions. However, in this case, it’s hard to estimate the impact of the methods on the overall results. In general, given that this paper will likely influence future generations of researchers, I wouldn’t like to see methods being reused in future that may be suboptimal.

Here, I highlight one main open issue, and suggest simple ways to address two other points. I would still like to express my enthusiasm for the overall approach though and hope the authors find the following useful.

Main issue:

Previously I highlighted the issue of reporting aggregate statistics across iterations/seizures/patients, with e.g. different numbers of seizures per patient. In my opinion, the statistically appropriate way to report the results is using hierarchical approaches that can account for the different nested levels of the data. Please report all statistics in this manner where applicable. Data visualisations should also acknowledge the nested nature of the data, where applicable. I realise this may not change your conclusions dramatically, but given the rising interest in subject-specific modelling, your paper will be setting an example and precedence in terms of the methods and techniques.

On the topic of statistical reporting, please also review the paper for accurate statistical reporting throughout. One example is “Difference was observed in terms of network modularity (U = 175, p = 0.004), but with modest effect size (mean modularity in the two groups 0.450 and 0.427).” Mean is probably not a useful measure of effect size here?

R3.6

“If the two nearest regions were in a similar distance from the midpoint (i.e.

satisfying d2/(d1 + 0.5mm) < 2, where d1 and d2 are the distances of the nearest and second

nearest region respectively), we did not assign the observation to any region, otherwise we

assigned it to the closest region. We opted for this cautious approach to minimize the risk of assigning the observation to the wrong source, guided by the rule that smaller amount of reliable data is preferable to larger amount of unreliable data.”

I previously asked you about a hypothetical scenario: “you wouldn’t assign a channel with d2=2 and d1=1 (2/(1+0.5)=4/3<2) to any region, but you would assign a channel with d2=20, d1=9 (20/(9+0.5)~=2.1>2) to the nearest region that is in this case 9 mm away?”

I still think you at least require a minimum distance before you apply your ratio criterion. Electrode contacts buried deep in white matter, far away from cortex are very unlikely to record a pure signal from a single region, regardless of what their ratio of d1 and d2 is. I think with your ratio method you are actually adding more unreliable data (in my example by including signals recorded far away from cortex). As a way forward, I’d suggest reporting the proportion of contacts that are e.g. more than 3mm away, and add a note to the methods and code that this should be re-visited in a future study.

R3.11

“To facilitate better generalization of the trained model to unseen seizures and to avoid the possibility of the training set being composed of large amount of similar seizures, we included at most two seizures from a single subject.” Maybe this is a misunderstanding, but by your logic, including one seizure per subject would be best, no?

Perhaps the easiest way forward is to simply state something like: “This resulted in the inclusion of xx patients with 2 seizures and yy patients with 1 seizure.”

**Have all data underlying the figures and results presented in the manuscript been provided?**

Reviewer #1: Yes

Reviewer #2: None

Reviewer #3: None

PLOS authors have the option to publish the peer review history of their article (what does this mean?). If published, this will include your full peer review and any attached files.

Reviewer #1: **Yes: **Mark A. Kramer

Reviewer #2: **Yes: **Marinho Antunes Lopes

Reviewer #3: **Yes: **Yujiang Wang
---

## [Decision Letter · Decision Letter 2]

10 Jan 2021

Dear Dr. Sip,

We are pleased to inform you that your manuscript 'Data-driven method to infer the seizure propagation patterns in an epileptic brain from intracranial electroencephalography' has been provisionally accepted for publication in PLOS Computational Biology.

Best regards,

Daniele Marinazzo

Deputy Editor

PLOS Computational Biology

Reviewer's Responses to Questions

**Comments to the Authors:**

Reviewer #3: The authors have now addressed my comments. Congratulation on this very interesting paper!

All the best & Happy New Year!

Yujiang

**Have all data underlying the figures and results presented in the manuscript been provided?**

Reviewer #3: None

PLOS authors have the option to publish the peer review history of their article (what does this mean?). If published, this will include your full peer review and any attached files.

Reviewer #3: **Yes: **Yujiang Wang

---

## [Editor Report · Acceptance letter]

11 Feb 2021

PCOMPBIOL-D-20-01581R2 

Data-driven method to infer the seizure propagation patterns in an epileptic brain from intracranial electroencephalography

Dear Dr Sip,

I am pleased to inform you that your manuscript has been formally accepted for publication in PLOS Computational Biology. Your manuscript is now with our production department and you will be notified of the publication date in due course.

With kind regards,

Alice Ellingham
